# Activation of JUN in fibroblasts promotes pro-fibrotic programme and modulates protective immunity

Lu Cui[1,7], Shih-Yu Chen[2,7], Tristan Lerbs[1], Jin-Wook Lee[3], Pablo Domizi [1], Sydney Gordon[4], Yong-hun Kim [1], Garry Nolan[5], Paola Betancur[6] & Gerlinde Wernig [1✉]

The transcription factor JUN is highly expressed in pulmonary fibrosis. Its induction in mice drives lung fibrosis, which is abrogated by administration of anti-CD47. Here, we use high-dimensional mass cytometry to profile protein expression and secretome of cells from patients with pulmonary fibrosis. We show that *JUN* is activated in fibrotic fibroblasts that expressed increased CD47 and PD-L1. Using ATAC-seq and ChIP-seq, we found that activation of *JUN* rendered promoters and enhancers of CD47 and PD-L1 accessible. We further detect increased IL-6 that amplified *JUN*-mediated CD47 enhancer activity and protein expression. Using an in vivo mouse model of fibrosis, we found two distinct mechanisms by which blocking IL-6, CD47 and PD-L1 reversed fibrosis, by increasing phagocytosis of pro-fibrotic fibroblasts and by eliminating suppressive effects on adaptive immunity. Our results identify specific immune mechanisms that promote fibrosis and suggest a therapeutic approach that could be used alongside conventional anti-fibrotics for pulmonary fibrosis.

[1] Department of Pathology, Institute of Stem Cell Biology and Regenerative Medicine (ISCBRM), Stanford University School of Medicine, Stanford 94305 CA, USA. [2] Institute of Biomedical Sciences, Academia Sinica, Taipei 11529, Taiwan. [3] Department of Genetics, Stanford University School of Medicine, Stanford 94305 CA, USA. [4] Orca Biosystems, 3475 Edison Way, Suite B, Menlo Park 94025 CA, USA. [5] Baxter Laboratories Department of Microbiology and Immunology, Stanford University School of Medicine, Stanford 94305 CA, USA. [6] Department of Radiation Oncology, University of California, San Francisco 94143 CA, USA. [7] These authors contributed equally: Lu Cui, Shih-Yu Chen. ✉email: gwernig@stanford.edu

With a 3-year survival rate of only 50%, pulmonary fibrosis has a terrible prognosis rivaling some of the worst malignancies. Pulmonary fibrosis is characterized by the spontaneous onset of progressive scarring of the lung in the absence of an infectious or autoimmune etiology[1–5]. Despite the discovery that germline mutations of TERT are highly prevalent in these patients, the pathophysiological mechanism of pulmonary fibrosis disease remains incompletely understood. There are no curative treatments other than lung transplantation, thus, novel therapies are desperately needed[6–10]. Clinical trials for nintedanib[11,12] and pirfenidone[13], two standard-of-care treatments targeting the receptor tyrosine kinases VEGFR, FGFR and PDGFR, and the TGFB pathways, respectively, are known to have a role in idiopathic pulmonary fibrosis, despite trends in improvement in mortality rates with antifibrotics patients keep progressing and a high unmet need to halt progression remains. Other treatment strategies including inhibiting toll-like receptors (TLR3, 4, 9) and metalloproteases, blocking macrophage activation and recruitment (mAbs TNFa and CCL2), targeting Th1 (=protective)/Th2(=profibrotic) imbalance with INFg and mAB IL-13, or immune modulatory treatments with CTLA4 and azathioprine in pulmonary fibrosis patients treated for cancer also failed to be effective in pulmonary fibrosis[14–18].

Fibroblasts are known to be at the core of the fibrotic response; however, quite surprisingly, they represent a poorly characterized cell type. Fibroblasts are heterogeneous, and no common consensus exists on their subtypes or biological properties such as signaling and plasticity. Recently, single-cell deconvolution of fibroblast heterogeneity was reported in a bleomycin-initiated pulmonary fibrosis mouse model[19]. However, no comprehensive single-cell data focusing on fibroblasts are yet available for pulmonary fibrosis patients. We tested the most-reported canonical fibroblast markers[20] and found that each only labeled a subset of fibroblasts. There are no universal fibroblast-specific markers. The level of heterogeneity with these so-called fibroblast markers could potentially bias analyses if only a single marker is selected to define fibroblasts. Furthermore, the mixed phenotype of certain fibroblasts like myofibroblasts, would also limit the extent of analyses if other mesenchymal markers were included for negative gating. Hence, we decided to define fibroblasts by only excluding leukocytes (CD45), epithelial cells (CK7) and endothelial cells (CD31). This strategy not only helped us to enrich but also to further characterize the heterogeneity of fibroblasts in pulmonary fibrosis. Monocytes and macrophages, as part of the innate immune response, are thought to have a critical role, regulating both injury and repair in various models of fibrosis. In addition, macrophage heterogeneity has emerged as an important area of study in fibrotic lung[21–24]. Adaptive immune processes have been shown to orchestrate existing fibrotic responses and various subsets of T cells have been shown to be enriched in fibrotic lung. In addition, increased activated regulatory T cells correlate with the severity of fibrosis[25–27]. Mass cytometry enables measurements of over 40 parameters simultaneously at the single-cell level when mass-tagged antibodies are used to label cellular proteins of interest and subsequently analyzed by time-of-flight mass spectrometry. Here, we used mass cytometry to characterize millions of primary lung cells from 11 pulmonary fibrosis patients and 3 normal donors at the single-cell level to determine the identities and functional aspects of various cell subsets in fibrotic lungs and systematically monitor interactions between cells in the microenvironment.

We previously showed that JUN caused severe lung fibrosis when induced in adult mice[1]. This represents a non-chemical, purely genetic model of lung fibrosis and highlights one critical transcription factor at the core of a general fibrotic response. Moreover, the activated phosphorylated form of JUN could represent a new biomarker to predict poor outcome in lung fibrosis. We observed that JUN induction in mice resulted in upregulation of the CD47 protein in fibroblasts within less than 24 h. CD47 is a key anti-phagocytic molecule that is known to render malignant cells resistant to programmed cell removal, or efferocytosis; it is a key driver of impaired cell removal[28,29]. We were then able to demonstrate that we could prevent fibrosis in mice with anti-CD47 immune treatment. Importantly, now we also find that anti-CD47 immune therapy largely reverses the fibrotic reaction. However, the molecular details of how JUN caused, or CD47 blockade disrupted, the development of lung fibrosis and the implications for human pulmonary fibrosis diseases remained unknown.

Here, our single-cell protein screening approach in fibrotic lung patients highlighted two immune regulatory pathways dysregulated in fibrotic lung, CD47 and PD-1/PD-L1. Antibody therapies against both are currently being tested in clinical trials for cancer and recently have also been demonstrated to prevent atherosclerosis[30–32]. In addition, we identified cytokine IL-6 at the core of progredient fibrosis in fibrotic lung. IL-6 is known to mediate its broad effects on immune cells (adaptive and innate) via a complicated signaling cascade in an almost hormone-like fashion, e.g., in vitro experiments demonstrated that lung macrophages produce soluble IL-6Ra, and that increased IL-6 signaling increased extracellular matrix production. A clinically tested blocking antibody against IL-6 is available and FDA approved for rheumatoid arthritis[33,34].

## Results

**PD-L1 and CD47 are upregulated in fibrotic fibroblasts.** To systematically profile the pathophysiology of human pulmonary fibrosis, we applied an -omics approach combining multiparameter single-cell mass cytometry and genome-wide chromatin accessibility assays together with a multiplexed Luminex secretome analysis as outlined in (Fig. 1a). For profiling with mass cytometry, single-cell suspensions of 14 representative lung samples, 11 fibrotic and 3 normal (all clinical information has been provided in Supplementary Table 1), were stained with a panel of 41 metal-conjugated antibodies (Supplementary Data 1) including 3 antibodies (CD45, CD31 and CK7) that allowed for manual gating of four distinct cell lineages: CD45[+] leukocytes, CK7[+] epithelial cells, CD31[+] endothelial cells and CD45[−]CK7[−]CD31[−] fibroblasts (Fig. 1b, gating strategy in Supplementary Fig. 7 and live cells counts in Supplementary Table 2). With this approach, we detected that the frequency of fibroblasts was 5-fold higher in fibrotic lungs (15% in normal lungs compared to 80% in fibrotic lungs), and leukocytes were 3-fold lower (60% normal compared to 20% in fibrotic lung). There was a mild but not significant decrease in epithelial cells and a negligible increase in endothelial cells (Fig. 1c). In addition to the increased abundance of fibroblasts, we performed a principal component analysis (PCA) of the expression level of all the markers (except the lineage markers CD45, CK7, CD31, CD61 and CD235a) on fibroblasts and demonstrated that fibrotic lung fibroblasts from the 11 fibrotic lung patients clustered together and were distinct from lung fibroblasts derived from normal lungs (Fig. 1d), suggesting fibroblasts in fibrotic lungs are not only increased in percentage but also differed phenotypically from control-lung fibroblasts. Consistent with the PCA results, viSNE plots showed enrichment of a distinct fibrotic lung-specific fibroblast subpopulation (Fig. 1e). Mass cytometry also demonstrated co-activation of phospho JUN and AKT in 50% of fibroblasts in un-manipulated human fibrotic lungs (Fig. 1f). The fibrotic lung-specific fibroblast subpopulation expressed high levels of CD47 and podoplanin, whereas PDGFRa, calreticulin and PD-L2 were moderately

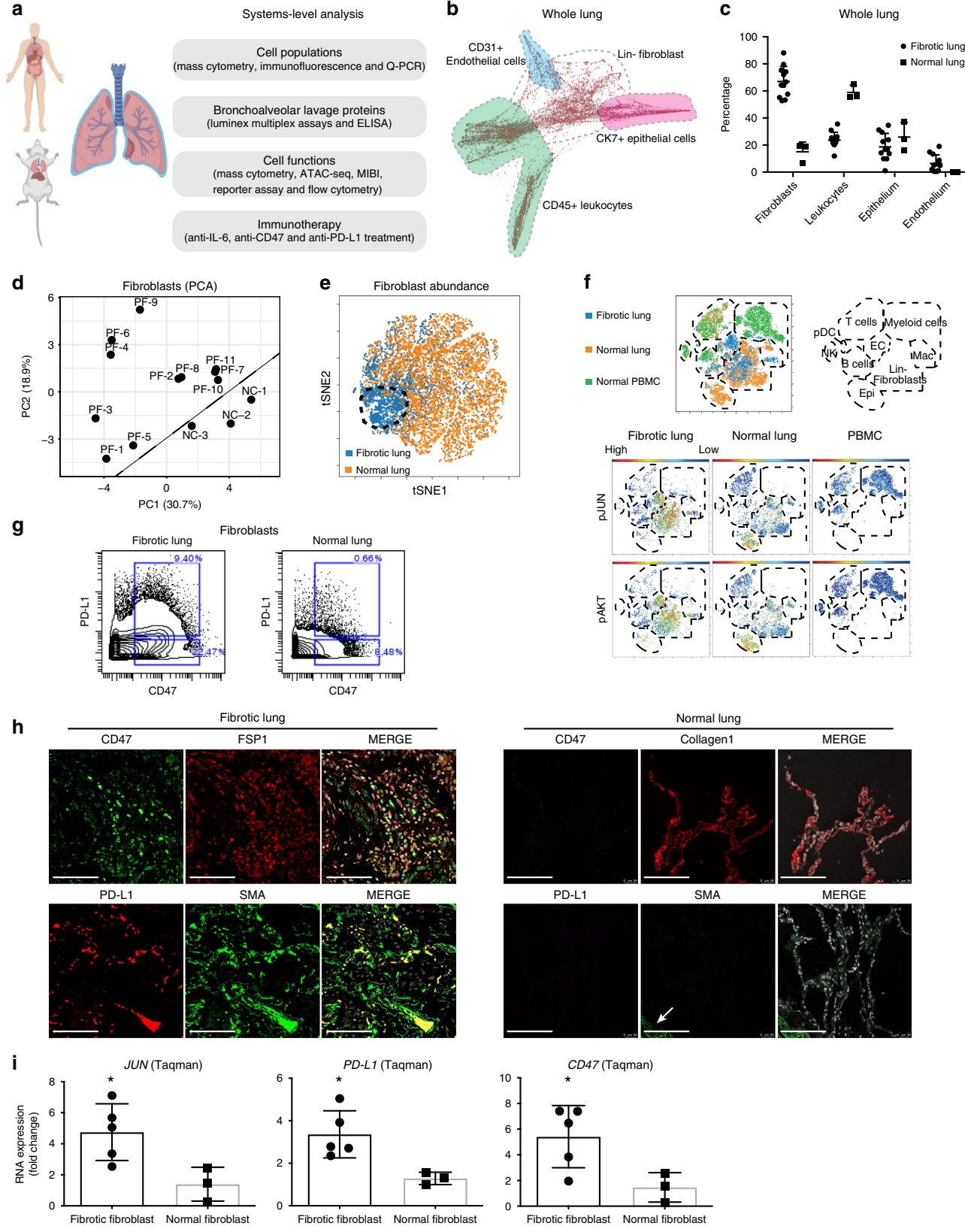

expressed (Supplementary Fig. 1a, b). As shown in Fig. 1g, >20% of the fibroblasts from fibrotic lungs expressed CD47 and a subset of ~10% co-expressed PD-L1. To assess the expression and distribution of these two immune-checkpoint proteins in intact lung tissues, we performed immune staining of fibrotic and normal control lungs. We detected abundant co-expression of CD47 with

FSP1, and PD-L1 with smooth muscle actin (SMA) in fibroblasts of fibrotic lung but not in normal lungs (Fig. 1h and Supplementary Fig. 1c showing the statistical analysis; Supplementary Fig. 1d showing the H&E stains of the same fibrotic lung and normal lung) by immune fluorescent stains and 9-color multiplexed ion beam imaging (MIBI), a technology which allows

**Fig. 1 Systems-level analysis of pulmonary fibrosis patients. a** Outline of our "Omics" approach in human fibrotic lung integrating proteomics, secretomics and genomics technology platforms to study the contribution of leukocytes and pathologic fibroblasts and to identify therapeutic targets. **b** Single-cell force-directed layout of fibrotic lung tissues. Shaded regions indicate the location of manually gated cell populations. **c** Frequencies of cell populations in the lung detected by mass cytometry (CyTOF). Data are displayed as mean ± SD of 11 fibrotic and 3 normal control-lung samples. **d** PCA was computed on fibroblast clusters from 11 individual pulmonary fibrosis patients (PF) and 3 normal donors (NC) mass cytometry datasets demonstrating that fibrotic and the normal fibroblasts were distinct from each other. **e** ViSNE maps of fibroblast mass cytometry data demonstrating that the abundance of fibroblasts differed. The data demonstrate a representative example per group and each point in the viSNE map represents an individual cell. **f** ViSNE analysis of mass cytometry data of fibrotic lung, normal lung and normal PBMCs revealed increased activation of the JUN and AKT pathways in fibrotic lung fibroblasts. Schematic diagram of the location of the indicated cell types on the viSNE map are based on the expression of lineage specific markers. Red indicates high and blue low protein expression. **g** Representative mass cytometry plots of the pro-fibrotic fibroblast population in fibrotic lung compared with normal lung. **h** Immune fluorescent stains confirmed increased CD47 and PD-L1 co-expression in lung fibroblasts from fibrotic lungs but not in normal controls (activated fibroblasts expressing FSP1+Collagen1+ and SMA+). The arrow indicates the blood vessel. (Scale bars, 100 μm). **i** RNA expression analysis of *JUN*, *PD-L1* and *CD47* in fibrotic and normal lung fibroblasts are detected by Taqman assay. Data are expressed as mean ± SD of 5 fibrotic fibroblasts and 3 normal fibroblasts and representative of at least three experiments. Data were analyzed by two-tailed unpaired *t*-test, *$P < 0.05$; **$P < 0.01$. See Supplementary Data 2 for statistical details. Source data are provided as a Source Data file.

concomitant staining of paraffin-embedded tissues with multiple antibodies and provides histologic resolution (Supplementary Fig. 1e) (all term definitions are in Supplementary Table 3).

We confirmed the upregulation of immune-checkpoint regulators at the gene-expression level with Taqman assay (Supplementary Table 4), where we detected increased *JUN*, *PD-L1* and *CD47* RNA expression in fibrotic over normal lungs (Fig. 1i). We also observed that secreted PD-L1 protein was increased in the bronchoalveolar lavages of fibrotic lungs but not normal lungs (Supplementary Fig. 1f). In conclusion, we found that fibroblasts in fibrotic lung are distinct from normal lung fibroblasts by both abundance and molecular phenotype, and a third of the fibroblasts upregulate either one of two immune-checkpoint proteins, CD47 and PD-L1, with dual upregulation demonstrated in ~9%.

**Macrophage and T cell exhibit immunosuppressive phenotype.** Although many different inflammatory cell subsets have been shown to have a role in fibrotic lung, their individual contributions to pulmonary fibrosis progression are unclear. Here we used the unbiased, comprehensive, single-cell characterization mass cytometry method to deeply investigate millions of leukocyte types contained in fibrotic and normal lungs. Among all CD45+ leukocytes contained in the lungs, we found quantitative differences of B, NK and dendritic cells (Supplementary Fig. 2a) in fibrotic compared to normal lung controls. We observed no significant differences in the percentage of T cells and macrophages (Fig. 2a, b).

For decades, investigations of the lung myeloid compartment have been mainly been limited to macrophages located within the airways, that is, alveolar macrophages which originate from fetal monocytes and are capable of self-renewal[35,36]. However, a number of recent reports have focused on the complexity of the myeloid cells present in the lung and provide evidence that interstitial macrophages are a heterogenous population comprising dendritic cells, tissue monocytes and nonalveolar macrophages[37]. By using all of the markers in the CYTOF panel (except the lineage markers CD45, CD31, CK7, CD61, CD235a, cPARP, CD3, CD4, CD8, CD19, CD56 and CD68), we noticed macrophages from fibrotic lungs were phenotypically different from normal lungs as demonstrated by PCA (Fig. 2c). Consistently, viSNE plots also showed the distinct distribution of macrophages from fibrotic lung patients (Fig. 2d; black dotted circle). In addition, markers enriched in alveolar macrophages (e.g., HLA-DR, CD169 and CD206) (Fig. 2e), as well as indoleamine 2,3-dioxygenase (IDO) (a rate-limiting enzyme in the metabolism of tryptophan which has a critical role in immune regulation) (Supplementary Fig. 2b, c) were downregulated in

fibrotic lung macrophages. Indeed, the ratio of interstitial macrophages (HLA-DR+CD206+CD169−, IM)[38] to alveolar macrophages (HLA-DR++CD206++CD169+, AM) was also significantly different between fibrotic and normal lungs (1.47 vs. 0.09, $P = 0.0079$) (Fig. 2f and Supplementary Fig. 2d). Consistent with the high degree of phenotypic heterogeneity in macrophages (Supplementary Fig. 2c), viSNE plots of manually gated alveolar and interstitial macrophages between fibrotic and normal lungs were also distinct, suggesting that development of unique AM and IM subtypes occurs in fibrotic lungs (Supplementary Fig. 2e). Meanwhile the macrophages in fibrotic lung tissues co-expressed PD-1 (Fig. 2g and Supplementary Fig. 2f showing the statistical analysis), a marker profile described for tumor-associated macrophages[39]. All these results suggest that the composition of macrophages changes markedly during the fibrotic process.

Next, we performed a thorough characterization of T cells. The percentage of T cells as well as CD4, CD8 and NKT cells did not differ between fibrotic and normal lungs (Supplementary Fig. 3a). However, an in depth analysis of T cells in pulmonary fibrosis demonstrated that specific subsets, such as naive CD4 and naive CD8 cells, were actually decreased (Fig. 3a, b), whereas no differences were detected for Th1, Th2 and Th17 T cells (Supplementary Fig. 3b). The most marked findings in fibrotic lung samples were the increased percentages of regulatory T cells (T$_{reg}$) and PD-1+CD4+T cells[40] (Fig. 3c, d and Supplementary Fig. 3c), as well as the increased percentages of exhausted T cells (Fig. 3e, f), which are suggestive of an immunosuppressive microenvironment. We confirmed these findings with immune stains and, indeed, found a greater percentage of T cells in the fibrotic lungs expressing PD-1. (Fig. 3g and Supplementary Fig. 3d showing the statistical analysis). We conclude that suppressive leukocyte types predominate in pulmonary fibrosis lungs.

**JUN controls pro-fibrotic and immune-checkpoint genes.** Given the tight correlation between *JUN* activation and immune-checkpoint-protein expression in the fibroblasts at the single-cell level, we speculated that JUN might directly regulate *CD47* and *PD-L1* at the transcriptional level in fibrosis. Previous work by Vierbuchen et al.[41,42] demonstrated that JUN, as part of the AP-1 (FOS/JUN) complex, can function as a pioneer transcription factor and acts as an enhancer selector to modulate the accessibility of DNA in fibroblasts. To that end, we conducted doxycycline (Dox)-inducible overexpression to investigate ectopic expression and CRISPR-editing to assay loss-of-function phenotypes with the goal of evaluating chromatin configuration in response to *JUN* expression using ATAC-seq (Assay for

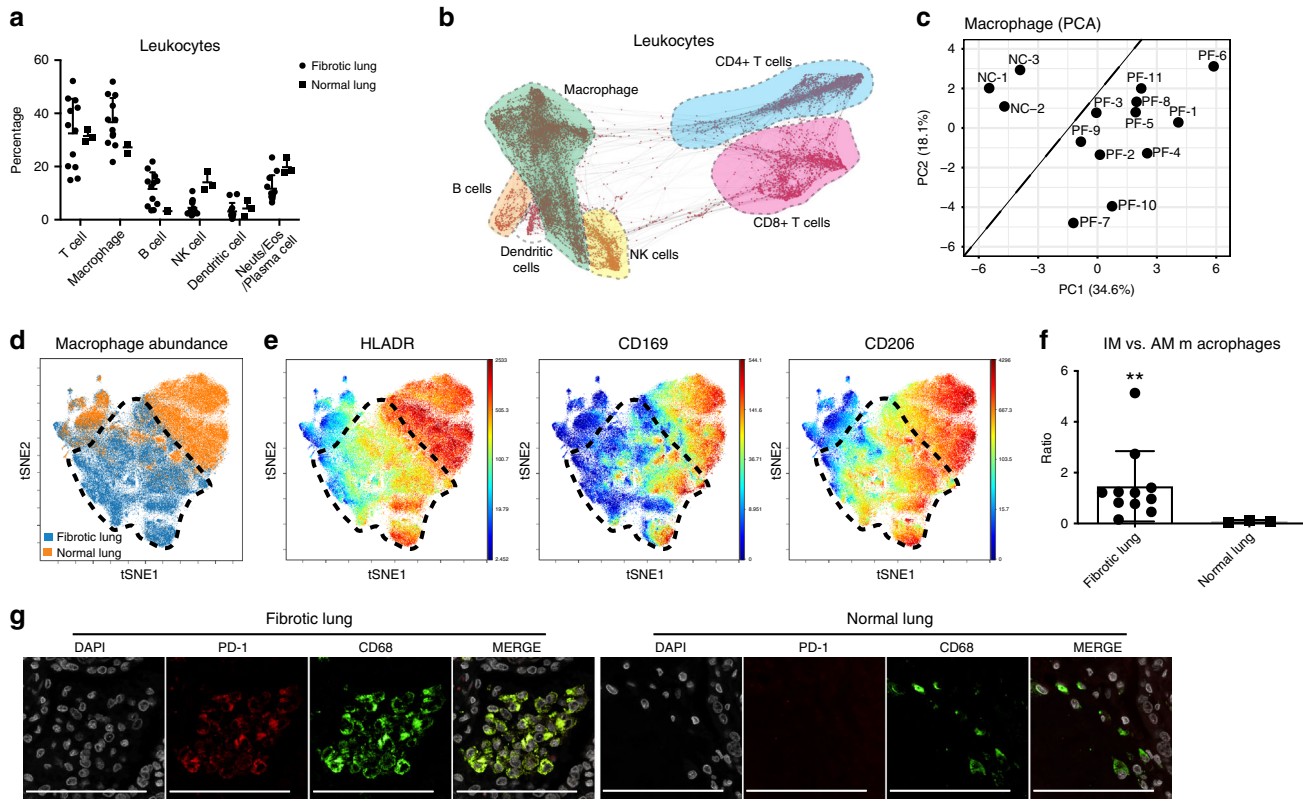

**Fig. 2 Lung fibrotic condition converts macrophages into an immunosuppressive phenotype. a** Main cluster frequencies of CD45$^+$ leukocytes (T cells, macrophages, B cells, NK cells, dendritic cells and other inflammatory cells such as neutrophils/eosinophils/plasma cells) contained in the lungs of pulmonary fibrosis patients and normal controls were quantified by mass cytometry. Data are expressed as mean ± SD of 11 fibrotic and 3 normal lung samples. **b** Computational analysis of mass cytometry data of leukocytes derived from fibrotic lungs with a single-cell, force-directed algorithm demonstrated that the different inflammatory subsets segregated as indicated on the map: CD4$^+$ T cells (blue), CD8$^+$ T cells (purple), macrophages (green), B cells (orange), NK cells (yellow) and the dendritic cell subset (white). **c** Principal component analysis (PCA) of manually gated macrophages (CD45$^+$ CD68$^+$ nonB nonT, nonNK live cells) indicating that macrophages derived from the pulmonary fibrosis lung (PF) clusters are distinct from those in normal lungs (NC). **d** A refined viSNE analysis of mass cytometry data demonstrating that macrophages derived from normal lungs (orange) have a distinct profile from fibrotic lungs (blue: black dotted circle). **e** ViSNE analysis of macrophages isolated from normal lungs and fibrotic lungs demonstrating decreased activation of HLA-DR, CD169 and CD206 expression in fibrotic lungs relative to controls. Each point represents a single cell, and the samples are color coded as indicated: blue colors represent low expression and yellow to red represent high protein expression. **f** The corresponding ratio of interstitial macrophages (HLA-DR$^+$CD206$^+$CD169$^-$, IM) versus alveolar macrophages (HLA-DR$^{++}$CD206$^{++}$CD169$^+$, AM) is displayed with mean ± SD of 11 fibrotic and 3 normal lung samples and analyzed by two-tailed unpaired *t*-test, **P < 0.01. **g** Representative images of immune fluorescent stains highlighted increased PD-1 expression in macrophages from fibrotic lung tissues (scale bars, 100 μm). See Supplementary Data 2 for statistical details. Source data are provided as a Source Data file.

Transposase-Accessible Chromatin using sequencing). This is a highly sensitive way to measure the chromatin accessibility of transposase with base-pair resolution genome-wide. We generated and performed ATAC-seq on primary fibrotic lung fibroblasts with (*JUN*-KO) or without (Control) genetic inactivation of *JUN* by CRISPR Cas9, followed by ATAC-seq and ChIP-seq on primary normal lung fibroblasts with (TetO-*JUN* Dox$^+$, *JUN*-OE) or without (TetO-*JUN* Dox$^-$) *JUN* overexpression. Hierarchical clustering of ATAC-seq signals revealed that the chromatin landscape of primary fibrotic lung fibroblasts was close to that of primary normal-lung fibroblasts with *JUN* activation; likewise, normal fibroblasts were clustered with fibrotic fibroblasts with *JUN* deactivation. Of the total open chromatin cis-regulatory elements across all samples, we found that the top differential peaks, enriched in proinflammatory genes such as *CD47*, *CD274* (PD-L1), *GLI1* and *NFKB1*, appeared to be regulated and downstream of *JUN* as their chromatin accessibilities decreased in *JUN*-KO fibrotic fibroblasts but increased in *JUN*-OE normal fibroblasts (Fig. 4a). *JUN* ChIP-seq data coupled with the ATAC data confirmed enrichment of bound JUN to the *JUN* promoter

region (shaded in red), which correlates with the more accessible chromatin state in overexpressed-*JUN* lung fibroblasts when compared to normal cells. This demonstrates that knockout of *JUN* decreases accessibility to its own promoter in a negative regulatory feedback fashion, whereas overexpression of *JUN* does the opposite. When we analyzed the DNA-bound JUN effects on the chromatin structure of *CD47* or *PD-L1*, we noticed that *JUN* enrichment (by *JUN* ChIP-seq) occurs preferentially in a distal genomic region (shaded in green; previously shown to be a super-enhancer of *CD47* in cancer)[43] for *CD47* and in the first intronic genomic region (shaded in green; reported as a *PD-L1* active enhancer)[44] for *PD-L1*, rather than in their corresponding promoters (shaded in red). The *JUN* enrichment observed in these two cases correlated with an increase in chromatin accessibility (detected by ATAC-seq) in lung-fibroblast cells compared to normal. This is particularly interesting as these changes are only present in our primary lung fibroblasts but not in any of the other previously published data on *JUN* ChIP-seq performed on cancer-cell lines such as A549, MCF-7, H1-hESC, HepG2 or K562. These results suggest that the binding of JUN to specific

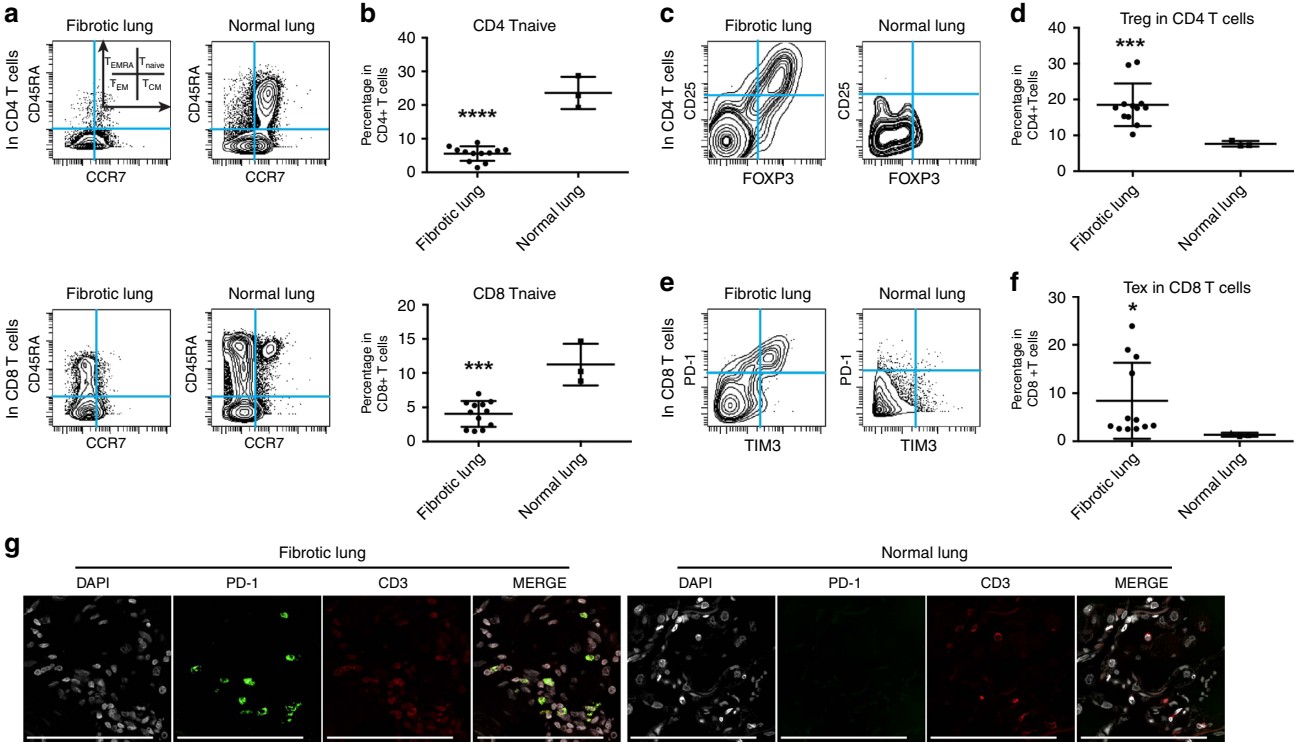

**Fig. 3 T cells present in fibrotic lungs mitigate a suppressive immune response in Lung fibrotic. a, b** Representative CyTOF plots and quantification of CD4$^+$ (top) and CD8$^+$ (bottom) naive T cells (CCR7$^+$ CD45RA$^+$) demonstrating decreased naive T cells in fibrotic lungs. Data represent mean ± SD of 11 fibrotic and 3 normal samples and are analyzed by two-tailed unpaired $t$-test, ***$P < 0.001$; ****$P < 0.0001$. **c, d** Representative CyTOF analysis showing increased frequency of regulatory CD4 T cells (T$_{reg}$: CD4$^+$ Foxp3$^+$ CD25$^+$) in fibrotic lungs. Data represent mean ± SD of 11 fibrotic and 3 normal samples and are analyzed by two-tailed unpaired $t$-test, ***$P < 0.001$. **e, f** Representative CyTOF plots and quantitative analysis indicating increased percentage of exhausted T cells (Tex: CD8$^+$ PD-1$^+$ TIM3$^+$) in fibrotic lungs. Data represent mean ± SD of 11 fibrotic and 3 normal samples and are analyzed by two-tailed unpaired $t$-test, *$P < 0.05$. **g** Representative images of immune fluorescent stains for PD-1 on T cells (CD3$^+$) highlighting increased percentage of PD-1$^+$ T cells in fibrotic lung samples (scale bars, 100 μm). See Supplementary Data 2 for statistical details. Source data are provided as a Source Data file.

*CD47* and *PDL1* regions might modulate accessibility to DNA in regulatory regions specific to fibrotic disease (Fig. 4b). We also compared the *JUN* KO against control and *JUN*-OE (TetO-*JUN* Dox$^+$) against control (TetO-*JUN* Dox$^-$) and found that chromatin peaks from promoters involved in the pro-fibrotic epithelial–mesenchymal transition, TGF-beta receptor signaling and Stat3 signaling pathways were most significantly modulated with *JUN* modification (Supplementary Fig. 4a, b). In addition, to demonstrate the physiological relevance of these findings, we compared our ATAC-seq data with published gene expression profiling from fibrotic and normal lungs[45] and found an overlap of 70 genes between the two datasets; among the most significant were genes encoding the pro-fibrotic epithelial–mesenchymal transition pathway, indicating that the JUN pathway could be a driver of fibrotic progression in pulmonary fibrosis (Supplementary Fig. 4c).

To demonstrate JUN regulation of *CD47* and *PD-L1*, we investigated their expression in fibroblasts following *JUN* modification after 4 days (Fig. 4c, d) and found that *JUN*-OE cells had higher levels of both RNA and protein expression. To confirm that changes in chromatin accessibility are pronounced in the enhancer region of *CD47*, we transduced fibrotic fibroblast cultures from patients with a construct containing the *CD47* enhancer followed by a GFP reporter such that the enhancer activity can be monitored by GFP expression (Fig. 4e). To study whether the JUN pathway regulates *CD47* through this enhancer, we serially measured GFP expression after *JUN* induction with a doxycycline-inducible system over a time course of 6 days and found that the *CD47* enhancer was active with *JUN* expression.

When we removed *JUN* from the culture by doxycycline withdrawal, we almost immediately lost *CD47* enhancer activity (Fig. 4f and Supplementary Fig. 4d). On the other hand, GFP expression is further downregulated in *JUN* KO cells (Fig. 4g and Supplementary Fig. 4d). Together, these findings demonstrated JUN is one of the key factors that can remodel chromatin and increase DNA accessibility to regulate the expression of fibrotic genes.

**IL-6 secreted by fibroblasts amplifies fibrosis via CD47.** To identify the cytokine pathways that cooperate with JUN, we profiled chemokines using a multiplex assay in the same bronchoalveolar lavage (BAL) human fibrotic lung samples that we analyzed by mass cytometry. Remarkably, we discovered that IL-6 was among the most highly upregulated cytokines compared to BALs from normal donors, along with the PDGF-BB growth factor and CCL5/CXCL5, both critical factors shown to be involved in connective tissue remodeling[46,47] (Fig. 5a). In addition, we detected increased IL-6 and family members in our Jun-induced pulmonary fibrosis mouse lung washings (Fig. 5b). To determine which cells were responsible for IL-6 secretion, we quantified IL-6 in the supernatant of cultures of lung fibroblasts, whole-bone marrow stroma and bone marrow-derived monocyte/ macrophages. We detected Jun-dependent IL-6 secretion only in the explanted lung fibroblast and marrow stroma cultures but not in bone marrow-derived monocytes/macrophage cultures which produced baseline IL-6 independent of Jun (Fig. 5c). These data indicated that IL-6 could be a critical downstream cytokine pathway involved in the Jun-mediated pro-fibrotic response.

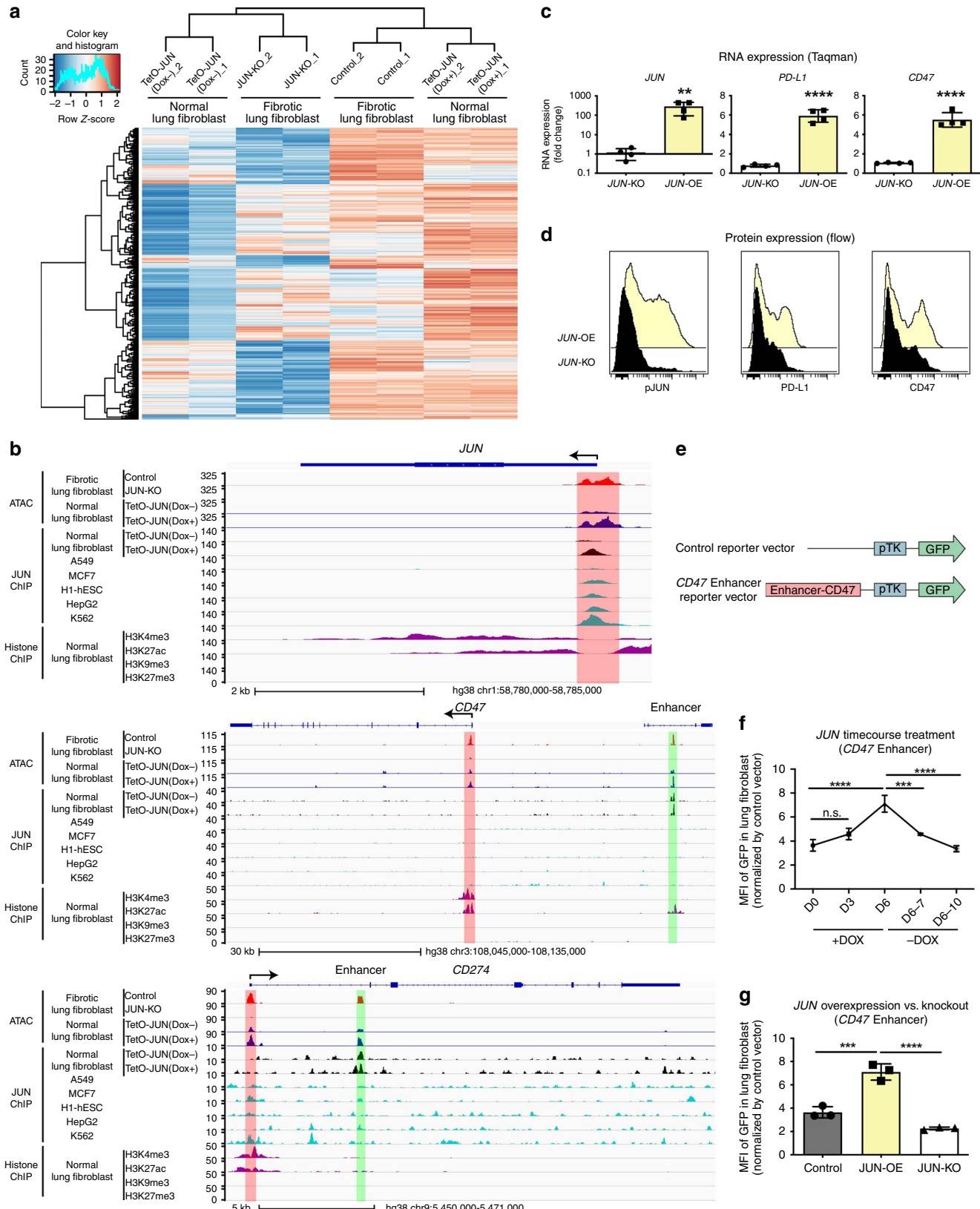

We subsequently evaluated the dependency of IL-6 on JUN in primary human-lung fibroblast cultures in which *JUN* was inducibly expressed or deleted with CRISPR Cas9. We found that promoter accessibility of *IL-6* and *IL-6R* and *IL-6ST* depended on JUN since their accessibility (shaded in red)[48] was lost with *JUN* deletion; in contrast, JUN's binding was highly enriched in the overexpressed-*JUN* lung fibroblast cells, indicating increased *IL-6* promoter accessibility in these cells (Supplementary Fig. 5a). IL-6 expression also correlated with JUN protein expression and, likewise, activation was lost with *JUN*

**Fig. 4 Accessibility of PD-L1 (CD274) and CD47 depended on JUN. a** Heatmap demonstrating dynamic chromatin changes in fibrotic lung fibroblasts with (*JUN*-KO) or without (Control) *JUN* deletion and normal lung fibroblasts with (TetO-*JUN* Dox⁺) or without (TetO-*JUN* Dox⁻) *JUN* activation.
**b** Representative genome browser tracks comparing ATAC-seq signal in fibrotic lung fibroblasts (with (*JUN*-KO) or without (Control) *JUN*-knockout) and also ChIP-seq signal in normal lung fibroblasts (with (TetO-*JUN* Dox⁺) or without (TetO-*JUN* Dox⁻) *JUN* overexpression) with A549, MCF-7, h1-hESC, HepG2 and K562 from published data at *JUN*, *CD47* and *CD274* loci. The red boxes highlight ATAC-seq and ChIP-seq peaks in the promoter sites of *JUN*, *CD47* and *CD274* (and enhancer is shown in green). We also compared our peaks with H3K4me3 or H3K27Ac (=histone mark for open chromatin), H3K9me3 or H3K27me3 (=histone mark for closed chromatin), ChIP-seq data generated from normal human-lung fibroblast is from published data, which highlighted the same areas respectively. **c** Gene expression changes in primary lung fibroblasts by comparing *JUN* knockout (KO) or overexpression (OE). Taqman assay were normalized to the value in *JUN* knockout. Data are expressed as mean ± SD from four experimental repeats. Two-tailed ratio paired *t*-test, **$P < 0.01$; ****$P < 0.0001$. **d** Representative flow cytometry histograms showing reduced expression of pJUN, PD-L1 and CD47 after *JUN* overexpression (OE) or knockout (KO). Yellow plot: *JUN* overexpression; Black plot: *JUN* knockout. **e** Vector maps of the control and *CD47* enhancer constructs used to engineer reporter cell lines. **f, g** *CD47* enhancer reporter assays demonstrating that doxycycline induced *JUN* expression initiated *CD47* enhancer expression, which disappeared when *JUN* expression was turned off (**f**) or *JUN* was knocked out (**g**). Data are expressed as mean ± SD from three independent experiments, Ordinary one-way ANOVA (Tukey's multiple comparisons test), n.s. non-significant; **$P < 0.01$; ***$P < 0.001$; ****$P < 0.0001$. See Supplementary Data 2 for statistical details. Source data are provided as a Source Data file.

deletion (Fig. 5d). Blocking the IL-6 pathway has been shown to attenuate lung fibrosis in mice and human fibrotic lung; however, how IL-6 contributes to the fibrotic process itself is unclear[34]. Our ELISA results demonstrate IL-6 concentrations of around 200 pg/ml in BAL of fibrotic lung patients (Supplementary Fig. 5b). Based on reported literature[49] and our own quantitative IL-6 measurements in normal lung BAL, baseline levels of IL-6 in healthy men are around 2–10 pg/ml. Hence, we treated normal lung fibroblasts with 1, 10, 100 ng/ml IL-6 to investigate the influence of IL-6 on the *CD47* enhancer. We found that increasing concentrations of IL-6 mimic JUN-mediated *CD47* enhancer activity (Fig. 5e and Supplementary Fig. 5c) and increased CD47 protein expression (Fig. 5f). IL-6 addition to *JUN* KO fibroblasts had no effect on *CD47* enhancer activity. Thus, IL-6 signaling cooperates with JUN to amplify JUN-mediated activation of the *CD47* enhancer in a synergistic manner.

**JUN induction in mouse recapitulates the key molecular events in human.** In this paper, we have made the intriguing observations that two critical immune checkpoint proteins-CD47 and PD-L1-are not only induced in a mouse model of lung fibrosis, but also in lung fibroblasts of human pulmonary fibrosis. These observations suggest that transcriptional programs active in fibrotic tissue try to dampen the immune response by blocking cytotoxic and phagocytic stimuli. Given these observations, we sought to determine whether blocking PD-1/PD-L1 would activate the immune system, both the innate and the adaptive, to eliminate overgrown fibroblasts in fibrotic lesions, and whether blocking different immune checkpoints simultaneously has additive antifibrotic effects and is well tolerated (Fig. 6). We previously established a *Jun*-inducible mouse model of lung fibrosis[1]. Here, we compared our genetic model to the frequently used bleomycin-induced model of lung fibrosis and found that the fibrotic response in the chemical-injury model was similar to our genetic lung-fibrosis model, i.e., similar distinct cell lineages [CD45⁺ leukocytes, EpCAM⁺ epithelial cells, CD31⁺ blood vessel endothelial cells and lineage negative fibroblasts (CD45-EpCAM-CD31-)] clustered together by X-shift clustering (Fig. 6a), and both resulted in activation of phospho JUN (Supplementary Fig. 6a).

**Immune checkpoints and IL-6 blockade clear fibrosis in mice.** In the same study, we also found that the PD-L1 is upregulated in murine lung fibrosis in a subset of pro-fibrotic fibroblasts (in both the genetic and bleomycin chemical-injury models), which highly co-expressed CD47. In addition to the expansion of macrophages, we also detected an immunosuppressive microenvironment as

was observed in human fibrotic lungs, including increased percentages of regulator T cells and exhausted T cells (Fig. 6b). We tested blockade of PD-1/PD-L1 (single/combined with anti-CD47 antibody) in the mouse model of lung fibrosis mediated by bleomycin in which we had initiated treatment at day 4 after fibrosis induction as a semi-therapeutic approach (Supplementary Fig. 6b). To assess and quantify the fibrotic response and the effects of immune-checkpoint inhibition in vivo over time, we performed serial high-resolution CT imaging of the lung weekly and found a striking reduction of the fibrosis in the lung (highlighted by reduced radio densities), a result most notable in lungs of mice treated with a combined IL-6, PD-L1 and CD47 blockade (Fig. 6c, d). In addition, we analyzed the mice using mass cytometry coupled with histopathological and immune stains of the lungs 2-weeks post-fibrosis initiation. Although there was increased PD-L1 co-expression with FSP1 in untreated mice with lung fibrosis, we found significantly decreased PD-L1⁺CD47⁺ fibroblasts. FSP1 has been described as a good marker for pro-fibrotic lung fibroblasts[50]. This finding correlates with a decreased collagen content of lung sections of the mice, which were treated with the triple combination of blocking antibodies against CD47 (Clone MIAP410) and IL-6 (Clone MP5-20F3), and an engineered non-antibody HAC protein, which was reported to be an effective anti-PD-L1 blockade in the treatment of mouse tumor models[39]. Similarly, B6.129S2-*Il6^{tm1Kopf}*/J (*IL-6* knockout) mice treated with anti-CD47 antibody and the PD-1 blocking reagent HAC similarly resolved their lung fibrosis confirming synergistic antifibrotic efficacy of IL-6 and immune-checkpoint inhibition (Fig. 6e and Supplementary Fig. 6c–e). To evaluate the effects of immune-checkpoint inhibition on innate immunity in fibrotic lung, we also used a humanized mouse model in which we have successfully engrafted primary human fibrotic lung fibroblasts in NOD-SCID gamma mice (NSG mice) underneath the kidney capsule. As key mediators of innate immunity, macrophages are the only remaining leukocytes in this NSG xenograft model and have been shown to interact with human PD-L1 via their PD-1. Based on luciferase detection of the human fibrotic fibroblast graft, we effectively resolved pathogenic pulmonary fibrosis by targeting PD-L1 with HAC protein. This outcome is confirmed by loss of GFP-positive grafted cells and lack of fibrosis by trichrome staining histology at the study endpoint (Supplementary Fig. 6f).

In summary, we demonstrated that immune-checkpoint CD47, PD-L1 and IL-6 signaling was markedly upregulated in a JUN-dependent fashion in pulmonary fibrosis patients and mouse lung-fibrosis models. We also showed that fibroblasts, macrophages and T cells in pro-fibrotic environments were all phenotypically different from the normal condition. Lastly, we

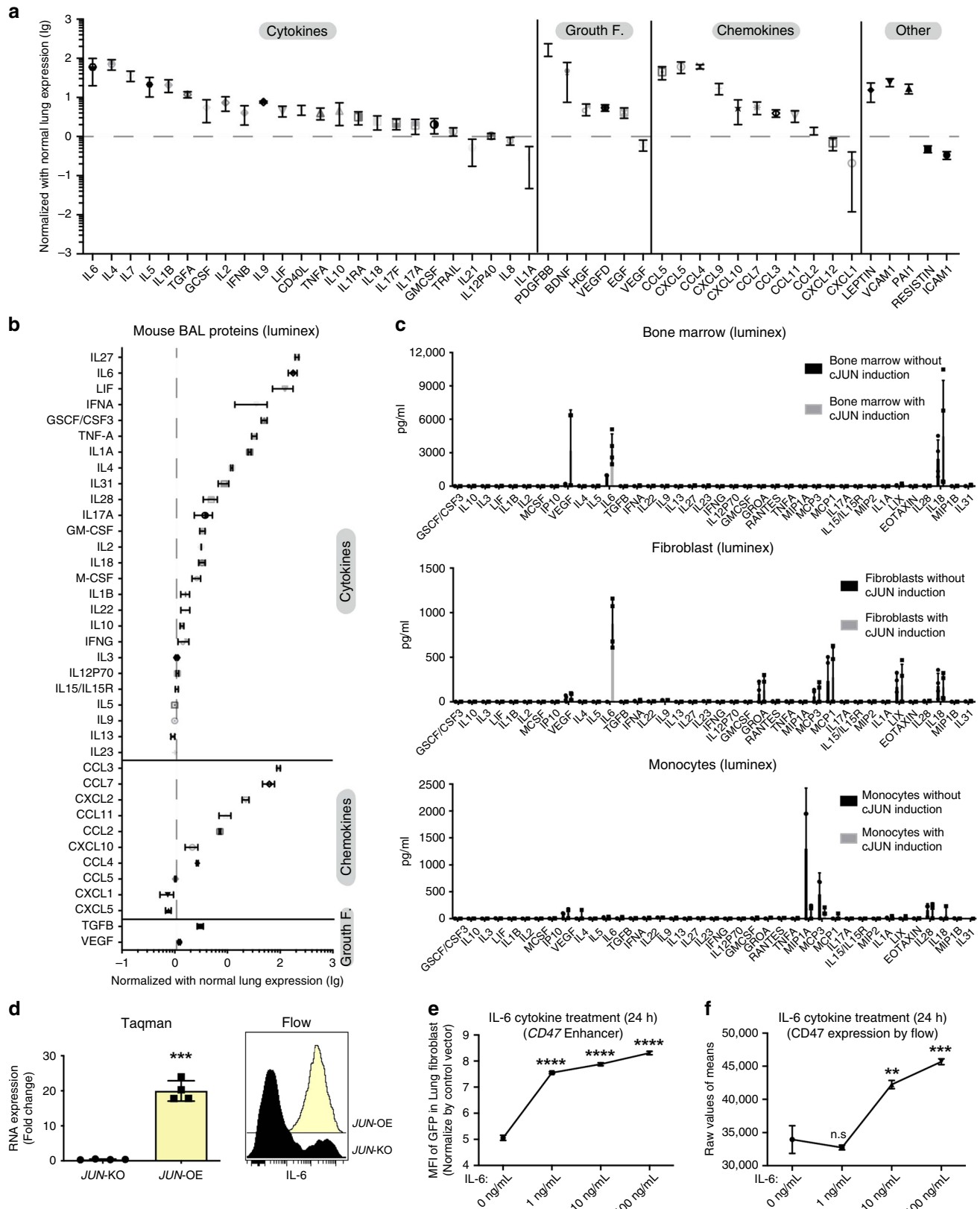

established that blockage of JUN-controlled pro-fibrotic and immune suppressive programs released immune suppression and hastened resolution of fibrosis in pulmonary fibrosis diseases (Fig. 7).

## Discussion

Fibrosis is a reactive process initiated by the stimulation of fibroblasts by leukocytes. The progression of fibrosis is determined by a dynamic balance between antifibrotic and pro-fibrotic

**Fig. 5 IL-6 mediates the pro-fibrotic response of JUN involvement. a** The secreted proteins in the lung bronchoalveolar lavage (BAL) of 4 fibrotic lung patients were quantified by Luminex assay, showing IL-6 as the highest expressed cytokine across all fibrotic patient BAL samples. Data were normalized by protein levels of the BAL of 3 normal lungs, and presented as mean ± SD. **b** Cytokines and chemokines in the fibrotic mouse bronchoalveolar lavage (BAL) after *JUN* induction were quantified by Luminex assay, and detected IL-6 was consistently among the most highly expressed cytokines in JUN-induced mouse fibrotic lungs indicative of IL-6-JAK-STAT pathway activation. Data were normalized by normal lung expression, and presented as mean ± SD, $n = 3$. **c** The cytokines/chemokines released from JUN-induced lung fibrotic mice-derived whole bone marrow ($n = 4$), fibroblasts ($n = 4$) and monocytes/macrophages ($n = 3$) in the medium after 48 h of Dox-initiated JUN induction were quantified by Luminex assay, demonstrating that whole bone marrow and fibroblasts are secreting increased IL-6 in response to JUN. Data were presented as mean ± SD. **d** Increased IL-6 expression levels were detected by Taqman assay and Flow cytometry in primary lung fibroblasts with *JUN* knockout (KO) or overexpression (OE). Data are expressed as mean ± SD from 4 experimental repeats. Two-tailed ratio paired *t*-test, ***$P < 0.001$. **e, f** IL-6 increased *CD47* enhancer activity at concentrations as low as 1 ng/ml (**e**) and protein expression at 10 ng/ml (**f**) in a dose-dependent fashion. Data are expressed as mean ± SD from three independent experiments, ordinary one-way ANOVA with multiple comparisons test, n.s. non-significant; **$P < 0.01$; ***$P < 0.001$. See Supplementary Data 2 for statistical details. Source data are provided as a Source Data file.

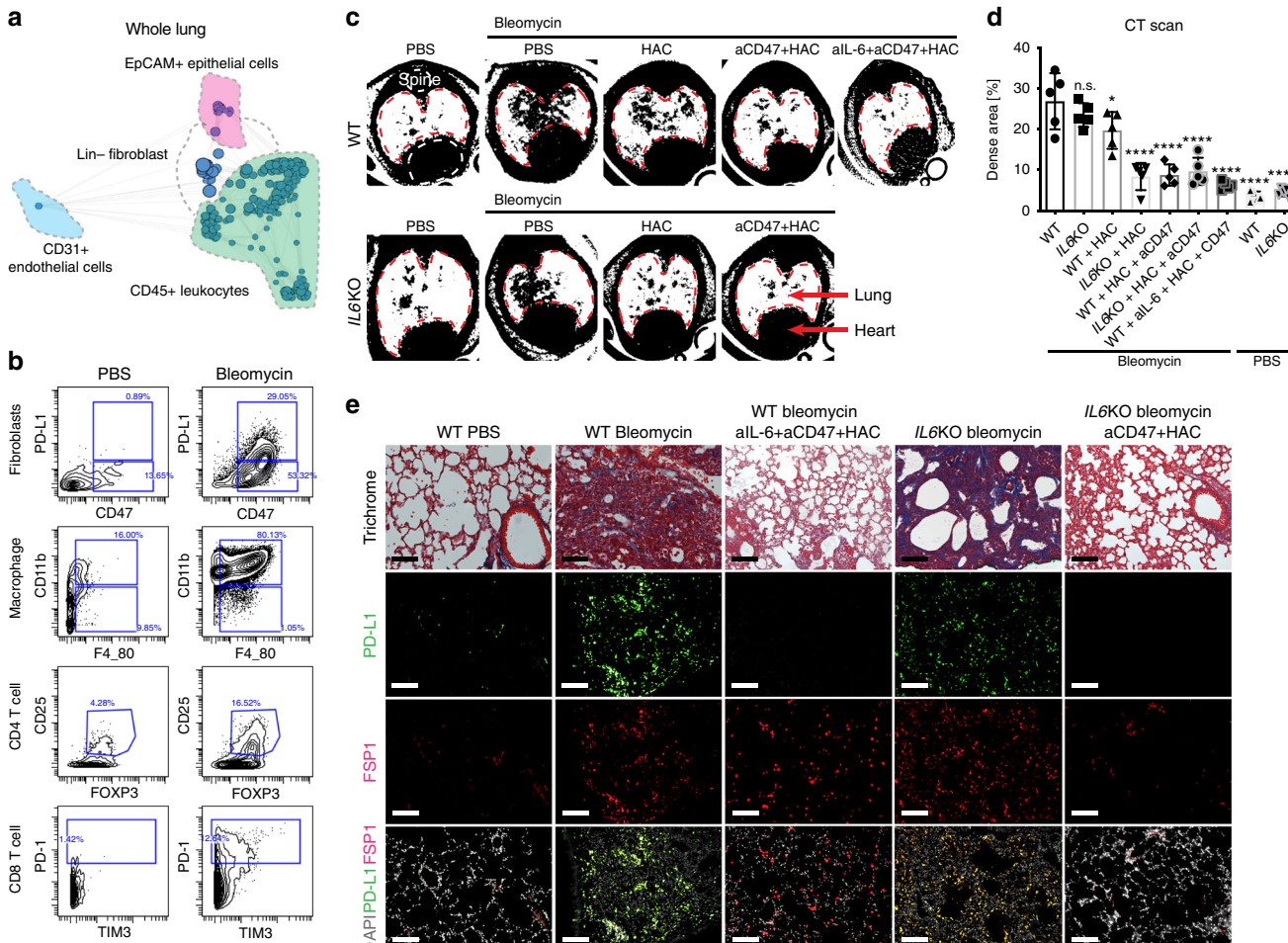

**Fig. 6 Inhibition of immune checkpoints together with IL-6 resolves lung fibrosis. a** Whole-lung scaffold map for Bleomycin-induced lung fibrosis in mice. Each node represents unsupervised cell clusters. **b** Representative mass cytometry plot demonstrating increased expression of immune-checkpoint proteins-CD47+ PD-L1+ in fibroblasts, an expansion of CD11b+ F4/80+ macrophages, regulator T cells (CD3+ CD4+ CD25+ FOXP3+) and exhausted T cells (CD3+ CD8+ PD-1+ TIM3+) in mouse model after fibrosis induction with bleomycin for 2 weeks. **c, d** Representative images of Micro CT scans of wildtype and B6.129S2-*Il6tm1Kopf*/J (IL-6KO) mice highlighting increased fibrosis in the lung after fibrosis induction (wildtype and IL-6KO mice) and much improved fibrosis after treatment with HAC (anti-PD-L1) alone or combined with a blocking antibody against CD47 or/and IL-6. Data are expressed as mean ± SD of five animals and analyzed by using one-way ANOVA followed by Tukey's multiple comparisons test for multiple comparison. n.s. non-significant; *$P < 0.05$; ****$P < 0.0001$. **e** Trichrome of lung sections of control mice, mice after fibrosis induction with bleomycin (wildtype and IL-6KO mice) and mice after treatment with blocking antibodies against IL-6 and CD47 and HAC (the blocking reagent against PD-L1) demonstrating markedly improved fibrosis (significantly decreased blue stained areas on Masson's trichrome stain which correspond to cross-linked collagen) and diminished PD-L1 expression in FSP1+ fibroblasts after treatment. Scale bar, 100 μm. See Supplementary Data 2 for statistical details. Source data are provided as a Source Data file.

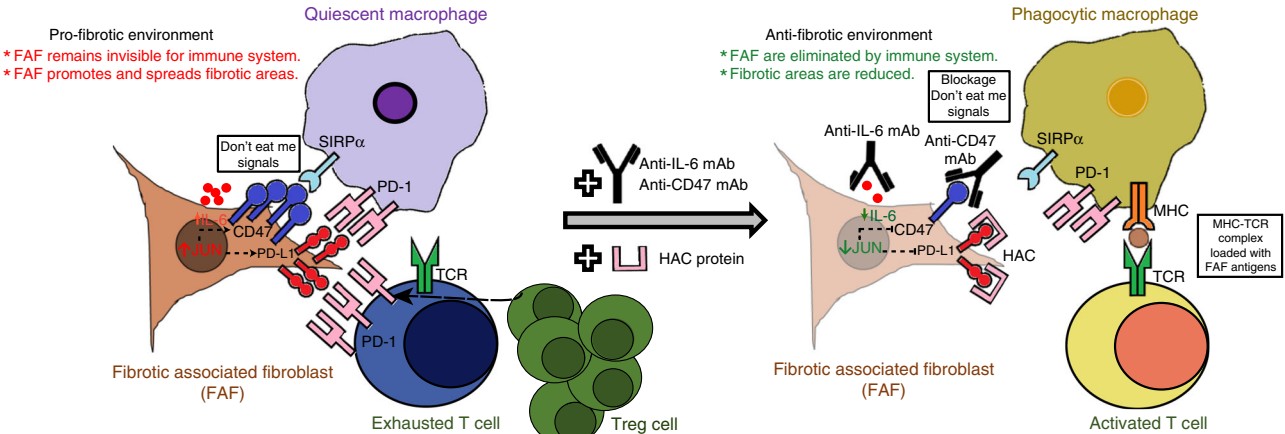

**Fig. 7 Schematic diagram of the proposed mechanisms of fibrosis clearance.** Left: In fibrotic lung, we find persistent myofibroblast activation in fibrotic plaques and JUN upregulation. JUN expression in fibrosis-associated fibroblasts (FAFs) appears to directly control the promoters and enhancers of *CD47* and *CD274* (PD-L1). The direct consequence is increased expression of these immune-checkpoint proteins in fibroblasts and dormant macrophages which do not phagocytose, but continue to release chronic inflammatory cytokines. JUN also directly regulates *IL-6* at the chromatin level. The increased expression and secretion of this potent cytokine leads to a suppressive adaptive immune response-chiefly T-cell exhaustion and upregulation of regulatory T cells. Right: Disrupting the suppression of the innate and adaptive immunity with CD47 and PD-L1 inhibitors as well as the proinflammatory IL-6 cytokine pathway stimulated phagocytic removal of pro-fibrotic fibroblasts and T-cell activation leading to clearance of the fibrosis in the lung.

mediators within a microenvironment composed of diverse cellular subtypes. Therefore, it is essential to not only identify the multiple cell populations simultaneously, especially fibroblasts and leukocytes, but also monitor their activities for generating a better understanding the pathophysiology of pulmonary fibrosis. Single-cell mass cytometry offers a number of significant advantages for profiling fibrotic disorders and complements studies using single-cell transcriptomic approaches, such as the recently published single-cell RNA-seq study by Misharin and his colleagues[51]. First, mass cytometry allows high throughput data collection in a single multiplexed experiment, thus enabling reliable comparisons of rare cell populations across different patients or conditions. Second, posttranslational modifications, which cannot be determined by transcriptomic analysis, are common mechanisms for regulating activities of proteins and can truly reflect their functional status inside cells. Finally, proteins but not mRNAs are the molecules, which mediate most crosstalk between cells through ligand and receptor interactions on the cell surface. As demonstrated in this study, we profiled millions of cells from the same cohort in a single multiplexed experiment and showed the heterogeneity of fibroblasts in pulmonary fibrosis patients. The upregulation of PD-L1 and CD47 in fibroblast subpopulations together with the immunosuppressive phenotype in T cells and macrophages suggested that the interactions between fibroblasts and leukocytes create a microenvironment in fibrotic lungs that restricts the removal of fibroblasts. The elevated activities of the JUN pathway in fibroblasts revealed by phospho-c-JUN-specific antibodies implicated the involvement of JUN pathways in the progression of pulmonary fibrosis. Indeed, by performing ATAC-seq, ChIP-seq and reporter assays, we demonstrated that the JUN pathway induces pro-fibrotic and immunosuppressive gene expression through a mechanism, which forms a positive feedback loop to increase CD47 expression via IL-6. Finally, the reduction of pulmonary fibrosis in mouse fibrosis models following blockade of IL-6, CD47 and PD-L1 validated the above mechanistic results, providing a potential therapeutic strategy to alleviate fibrosis in pulmonary fibrosis patients.

JUN is a transcription factor that coordinates the transcriptional regulation of genes that are essential for cellular growth and proliferation, such as the cell cycle, self-renewal, metabolism and survival. Here, we demonstrated that *JUN* expression in fibroblasts, contributes to fibrotic disease not only by directly increasing activity through cell-intrinsic, pro-fibrotic programs, but also by influencing the host's overall immune response. We showed that *JUN* expression in fibroblasts increases IL-6 expression and secretion, which has direct effects on both the adaptive and innate immune systems. Furthermore, we discovered that *JUN* expression in fibroblasts upregulates the expression of the immune-checkpoint genes *CD274* (PD-L1) and *CD47*. Thus, the overexpression of *JUN* may be a general mechanism by which fibroblasts initiate the intrinsic fibrosis program and maintain the progression of fibrosis non-cell autonomously via interacting with the immune system. Interestingly, the dual effects of transcription factors as the key drivers of pathophysiology might be a general principle applicable to many disorders as a similar mechanism has been shown for MYC in cancer[52].

Clinically, CD47 and PD-L1 are of interest because excellent reagents have already been developed by multiple pharmaceutical companies to target both immune-checkpoint molecules. Antibody therapies against both are currently being tested in clinical trials for cancer[31,32,39,40], and a blocking antibody against IL-6 is FDA approved to treat rheumatoid arthritis and acute cytokine release syndrome, a side effect of CAR-T cell therapy[33]. Therefore, extensive safety information will become available from these cancer studies. However, so far the focus has been on treating cancer; application of these immune therapies to other diseases has not been sufficiently been explored. Our work is the first to target several immune checkpoint and immune regulatory proteins in combination to achieve synergy in disrupting pro-fibrotic pathways in preclinical mouse models of fibrosis in a semi-therapeutic approach. In addition, our in vivo studies confirm our mechanistic studies, which are conceptually novel and demonstrate for the first time that *JUN* activation drives the expression of CD47 and PD-L1 in 30% of fibrotic lung fibroblasts and is mediated by IL-6 signaling. This finding will be of major importance to the design of future antifibrotic therapies based on immune regulatory proteins. Also, JUN inhibition (once systemic toxicity issues have been solved) may be by targeted delivery[53]. Given the excitement and seemingly high success of targeting those two immune-checkpoint pathways in cancer, clinical

development for a different application like fibrosis will be greatly facilitated by piggybacking on the experience in cancer patients. Although blocking antibodies against CD47 and IL-6 are relatively safe, clinical experience with PD-L1 inhibitors in cancer demonstrate severe pulmonary side effects potentially limiting their use and warranting caution in pulmonary fibrosis patients.

In conclusion, our data suggest that inhibition of each of these pathways (single/combined) in combination with standard of care therapy could potentially be used as a novel therapeutic approach for pulmonary fibrosis diseases. The consequences of our study are significant because our data represent a critical preclinical study that may ultimately be the basis for IND-enabling studies toward the goal of halting or even reversing the often-fatal course of pulmonary fibrosis diseases.

## Methods

**Isolation of fibroblasts from human tissue**. Human fibroblasts were obtained from discarded fresh lung tissues from de-identified patients. The tissue was minced, filtered through 70-μm filters, and then centrifuged at $600 \times g$ for 5 min to remove non-homogenized pieces of tissue. The tissue homogenate was treated with ACK lysing buffer (ThermoFisher) for 10–15 min, centrifuged at $600 \times g$, washed twice in DMEM with 10% fetal bovine serum (Gibco), plated at a density of approximately 500,000 cells/cm$^2$ in DMEM with 10% fetal bovine serum, 1% penicillin/streptomycin (ThermoFisher Scientific) and Ciprofloxacin (10 μg/ml, Corning) and kept in an incubator at 37 °C 95% $O_2$/5% $CO_2$. Media was changed after 24 h and cells were cultured until 80–90% confluent before each passage.

**Single-cell mass cytometry (CyTOF)**. Samples were processed as described[1]. Briefly the cell samples were fixed with 2% paraformaldehyde at room temperature for 20 min followed by two washes with PBS containing 0.5% BSA. Formaldehyde-fixed cell samples were incubated with metal-conjugated antibodies against surface markers for 1 h, washed once with PBS containing 0.5% BSA, permeabilized with methanol on ice for 15 min, washed twice with PBS containing 0.5% BSA and then incubated with metal-conjugated antibodies against intracellular molecules for 1 h. Cells were washed once with PBS containing 0.5% BSA, and then incubated at room temperature for 20 min with an iridium-containing DNA intercalator (Fluidigm) in PBS containing 2% paraformaldehyde. After intercalation/fixation, the cell samples were washed once with PBS containing 0.5% BSA and twice with water before measurement on a CyTOF mass cytometer (Fluidigm). Normalization for detector sensitivity was performed as previously described[54]. After measurement and normalization, the individual files were analyzed by first gating out doublets, debris and dead cells based on cell length, DNA content and cisplatin staining. ViSNE maps were generated with software tools available at https://www.cytobank.org by considering all surface markers.

**Immunostaining**. Tissue sections (4-μm-thickness) for immunofluorescence staining were cut from tissue blocks of archival de-identified human biopsies using a microtome. The sections were baked at 65 °C for 20 min, deparaffinized in xylene and rehydrated via a graded ethanol series. The sections were then immersed in epitope retrieval buffer (10 mM sodium citrate, pH 6) and placed in a pressure cooker for 45 min. The sections were subsequently rinsed twice with dH$_2$O and once with wash buffer (TBS, 0.1% Tween, pH 7.2). Residual buffer was removed by gently touching the surface with a lint-free tissue before incubating with blocking buffer for 30 min. Blocking buffer was subsequently removed, and the sections were stained overnight at 4 °C in a humidified chamber. The following morning, the sections were rinsed twice in wash buffer, a secondary antibody (Invitrogen, Carlsbad, CA) was used for visualization of signal. Images of histological slides were obtained on a Leica Eclipse E400 microscope (Leica, Wetzlar, Germany) equipped with a SPOT RT color digital camera model 2.1.1 (Diagnostic Instruments, Sterling Heights, MI). For MIBI, slides were postfixed for 5 min (PBS, 2% glutaraldehyde), rinsed in dH$_2$O and stained with Hematoxylin for 10 s. At the end, the sections were dehydrated via a graded ethanol series and air dried using a vacuum desiccator for at least 24 h before imaging. MIBI imaging are performed by NanoSIMS 50L spectroscopy (Cameca, France) at Stanford Nano Shared Facilities (SNSF) and analyzed by using Image with Plugin OpenMIMS (NRIMS, http://www.nrims.hms.harvard.edu).

**ELISA**. Bronchoalveolar lavages were harvested from patient lungs immediately (within 5 min) after explant. Five ml were injected into the peripheral airspaces and at least 2 ml harvested for all specimens, which were subsequently snap-frozen in liquid $N_2$. All specimens were surgical specimens and no post-mortem specimens were included. The expression of PD-L1 and IL-6 from bronchoalveolar lavages was quantitated following the protocols of ELISA kits: Human IL-6 Quantikine ELISA Kit and Human/Cynomolgus Monkey B7-H1/PD-L1 Quantikine ELISA Kit from R&D Systems.

**Lentivirus preparation**. Around 80–90% confluent 293T cells were transfected with 4 μg transfer plasmid (*JUN* tet-on overexpression plasmid, tetracycline-controllable transactivator plasmid, *JUN* CRISPR knockout plasmid, TK control reporter plasmid, E7TK *CD47* enhancer reporter plasmid and Luciferase-GFP plasmid), 2 μg pRRE Packing plasmid (GAG and Pol genes), 1 μg pRSV Packing plasmid (Rev gene), 1 μg pMD2.G enveloping plasmid and 24 μg PEI. The day after transfection, cell media was replaced, and cells were incubated for further 48 h, with media collection and replacement every 24 h twice. Cell media was centrifuged at $600 \times g$ for 10 min at 4 °C. Then, the supernatant was filtered through a 0.22-μm strainer, ultra-centrifuged at $25,000 \times g$ for 2 h, aliquoted and flash-frozen.

**CRISPR-mediated genome engineering**. Following literature protocols[55,56], the sequences of 2 site-specific guide RNAs (sgRNAs) that target exon1 of the *JUN* gene were selected using the CRISPR Design Tool 43. Oligonucleotides with these sequences were cloned into the lentiCRISPRv2 vector (Addgene, Cambridge, MA), which uses Puromycin selection to enrich for cells with *JUN* knockout.

The sgRNA sequences:

| | |
|---|---|
| *JUN* sgRNA_1 F | CACCGTGAACCTGGCCGACCCAGTG |
| *JUN* sgRNA_1 R | AAACCACTGGGTCGGCCAGGTTCAC |
| *JUN* sgRNA_2 F | CACCGCCGTCCGAGAGCGGACCTTA |
| *JUN* sgRNA_2 R | AAACTAAGGTCCGCTCTCGGACGGC |

**Doxycycline (DOX) inducible *JUN* overexpression**. To generate the *JUN* tet-on overexpression plasmid, we cloned *JUN* cDNA into IRES-Hygro-TetO-FUW vector (Addgene, Cambridge, MA). To generate doxycycline (Dox) inducible *JUN* overexpression, the tetracycline-controllable transactivator (rtTA) lentivirus was infected with the *JUN* tet-on overexpression plasmid. Dox (2 mg/ml) was applied during the infection to turn on *JUN* overexpression. Hygromycin selection was started on the second day for 2 days.

**ATAC-seq and ChIP-seq library preparation and sequencing library preparation**. The primary fibrotic lung fibroblasts were infected with *JUN* knockout lentiviruses, followed by 4 days of puromycin selection; meanwhile the normal lung fibroblasts were infected with *JUN* overexpression lentiviruses, followed with 2 days hygromycin selection. As described in detail in the published protocol[57], a transposition reaction was initiated in each sample containing 50,000 nuclei as assessed by counting. Subsequently, a DNA library was prepared using a Nextera DNA Library Preparation Kit (Illumina) and sequenced on the Illumina Nextseq500 platform with 75-bp × 2 paired-end reads. ChIPs and their respective inputs were generated as previously described[43]. The libraries were prepared by using a TruSeq ChIP sample preparation kit (Illumina) and sequenced by Nextseq500 pair-end sequencing (75 bp).

**Deep sequencing data analysis**. ATAC-seq and ChIP-seq data analysis used the Kundaje lab pipeline with the following tools and versions: Cutadapt v1.9.1, Picard v1.126, Bowtie2 v2.2.8, MACS2 v2.1.0.20150731 and Bedtools v2.26. First, Nextera adaptor sequences were trimmed from the reads by using Cutadapt program v1.9.1. These reads were aligned to human genome hg38 using Bowtie2. The standard default settings were modified to allow mapped paired-end fragments up to 2 kb. Only the reads with mapping quality greater than 30 were kept, and the duplicated reads were removed using Picard tools v1.126. The reads from mitochondria were also removed, then we converted PE BAM to tagAlign (BED 3 + 3 format) using Bedtools v2.26 functions. Differential expression analysis were done by DESeq2 (http://bioconductor.org/packages/release/bioc/vignettes/DESeq2/inst/doc/DESeq2.html). Differential peaks had a *P*-value < 0.01 and absolute log2 fold change above 1.

**Flow cytometry**. The analysis of IL-6 or surface molecules (CD47 and PD-1) was performed using monoclonal antibodies listed in the Supplementary Data 1. Data were acquired by LSRII or LSRFortessa flow cytometers and analyzed using FlowJo software or Cytobank.

**Bleomycin-induced mouse model and JUN-induced lung fibrosis mouse model**. For bleomycin administration, mice were anaesthetized with isoflurane followed by intratracheal instillation of bleomycin (4 U/kg per body weight) in 100 μl PBS as previously described[58]. As published previously[1], the reverse tetracycline transactivator (rtTA) ubiquitously inducible *Jun* was expressed using the *Rosa26* promoter, and the inducible cassette was targeted downstream of *Col1a1* promotor. After crossing, genotyping of the inducible JUN mice was performed using primers for the transgene *JUN* and Rosa26. *JUN* under the *Col1a1* promotor was induced by adding doxycycline (2 mg/ml) (Millipore Sigma) to the drinking water.

**In vivo antibody blockade**. For CD47 antibody blockade experiments, mice were injected intraperitoneally (IP) with a dose of 500 μg CD47 antibody (Clone MIAP410, Bioxcell) diluted in 100 μl of PBS on day 4. The same dose was then given every other day up to two weeks. For PD-L1 blockade experiments, HAC protein (250 μg, IP) was given daily for the entire treatment period. For IL-6 antibody blockade experiments, mice were injected intraperitoneally with 20 mg/kg dose of an anti-IL-6 monoclonal (Clone MP5-20F3, Bioxcell) antibodies twice a week for 2 weeks.

**CT scan**. Mice were anesthetized, and CT scans were performed using a Bruker Skycan 1276 (Bruker, Belgium). CT scans were then analyzed with Bruker Skycan tools. To determine dense areas within the lungs, binary pictures were created using the heart as the cutoff value to split the tissue in dark (having at the least the same density as the heart) and white areas. Representative total lung and dense areas were measured in the upper and middle field. Finally, the fraction of the dense areas within the total lung areas was calculated.

**Kidney capsule transplantation**. After mice had been anesthetized, the areas over the right and/or left flank were shaved and disinfected. Thereafter, a flank cut was made. Subcutaneous tissues were bluntly removed and an incision into the abdominal cavity was made. The kidney was luxated out of the abdominal cavity and a slight incision into the renal capsule was made. The renal capsule was bluntly detached from the renal tissue and $2 \times 10^5$ cells suspended in 10 μl of matrigel were injected under the kidney capsule. The kidney was pushed back into the abdominal cavity. The abdominal cavity and skin were closed using sutures.

**Luciferase-based optimal imaging (BLI)**. 100 μl of luciferin substrate was intraperitoneally injected. Fifteen minutes later, optical imaging was performed using a Lago optical imaging system (Spectral imaging instruments, AZ, USA). Analysis was done with the Aura Software from the same manufacturer.

**Statistics**. Statistical analyses were performed using Prism software (GraphPad Software). Statistical significance was determined by the unpaired Student's t-test for comparisons between two groups and one-way ANOVA for multigroup comparisons (n.s. non-significant; $P > 0.05$; *$P < 0.05$; **$P < 0.01$; ***$P < 0.001$; ****$P < 0.0001$). In statistical graphs, points indicate individual samples, and results represent the mean ± SD unless indicated otherwise.

**Reporting summary**. Further information on research design is available in the Nature Research Reporting Summary linked to this article.

## Data availability

Raw ATAC-seq and ChIP-seq data have been deposited in the Gene Expression Omnibus (GEO) database under accession code GSE114844. JUN ChIP-seq of HepG2 (GSM935364), MCF-7 (GSE91550), H1-hESC (GSM935614), A549 (GSE92221) and K562 (GSM1003609), Histone ChIP-seq data of H3K4me3 (GSM733723), H3K27ac (GSM733646), H3K9me3 (GSM1003531) and H3K27me3 (GSM733764) and RNA-seq data of fibrotic lungs (GSE52463) are from the public GEO database. The data that support the findings of this study are available from the corresponding author upon reasonable request. The source data underlying Figs. 1c, i, 2a, f, 3b, d, f, 4c, f, g, 5a–f and 6d and Supplementary Figs. 1c, e, f, 2b, f, 3a, c, d, 5b and 6e, f are provided as a Source Data file.

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

## Acknowledgements

We thank: Dr. Irving Weissman for insightful discussions; Dr. Tushar Desai for providing access to pulmonary fibrosis patient samples; Drs. Tushar Desai and Mark Nicolls for helpful discussions; Diane Durnam for English editing services; the Stanford Nano Shared Facilities (SNSF) for help with MIBI imaging; and the Human Immune Monitoring Center at Stanford for cytokine and chemokine Luminex quantifications. Our studies were supported by National Institute of Health, Heart Lung and Blood (NHLBI), the Scleroderma Research Foundation and Boehringer-Ingelheim Pharmaceuticals, Inc. (BIPI). BIPI had no role in the design, analysis or interpretation of the results in this study; BIPI was given the opportunity to review the manuscript for medical and scientific accuracy as it relates to BIPI substances, as well as intellectual property considerations. T.L. is supported by the Deutsche Forschungsgesellschaft (DFG).

## Author contributions

Conceptualization, G.W.; Methodology, L.C. and S.-Y.C.; Investigation, G.W. L.C., S.-Y.C., T.L., P.D., Y.-H.K.; Writing – original draft, G.W.; Writing – review and editing, G.W., L.C.; Funding acquisition, G.W.; Resources, J.-W.L., S.G., G.N., P.B.; Supervision, G.W.

## Competing interests

The authors declare no competing interests.

## Study approval

De-identified patient specimens in paraffin and discarded fresh patient tissues were used for our studies as approved by the institutional review board at Stanford University, IRB-18891 and IRB-39881. Written, informed consent was obtained from the patient prior to surgery. Tissue was collected in the operating room, placed directly into sterile saline, and kept on ice for transport. We received primary lung tissues exclusively from patients with end-stage pulmonary fibrosis undergoing transplantation. Therefore, our cohort of patients represents severely fibrotic lung disease. It has been challenging to receive normal control-lung tissues. Although we have received lung tissues from normal lung resections from tumor resections from Stanford tissue bank as well as lungs from rapid autopsies, it appeared that the only normal lung tissues harvested during surgery by the tissue bank were of sufficient viability to include in our CyTOF studies; while other cell-type fractions appeared representative. We noted a bias toward less endothelial cells in the normal biopsies due to the relatively small amounts of lung tissue we received from excisional biopsies from the tissue bank. Animal studies were approved by the Stanford University Administrative Panel on Laboratory Animal Care. Mice were maintained in Stanford University Laboratory Animal Facility in accordance with Stanford Animal Care and Use Committee and National Institutes of Health guidelines (SU-APLAC 30911 and SU-APLAC 30912).
