## [Peer Review File · Nature Communications]

Reviewers' comments:

Reviewer #1 (Remarks to the Author):

Peer review of the manuscript with the number NCOMMS-19-12042 at Nature Communications by Cui L, Chen SY et al and with the title: "Activation of JUN in fibroblasts promotes pro-fibrotic programme and dampens protective immunity"

The authors characterized the heterogeneous population of cells that is relevant for the pathophysiology of human idiopathic pulmonary fibrosis. The authors analyzed single cell suspensions of 14 representative lung samples (11 IPF samples and 3 control samples) by mass cytometry after staining with a panel of 41 metal conjugated antibodies. The author selected the antibodies used for the mass cytometry analysis based on published literature. The authors claim to have selected antibodies specific against the most published canonical fibroblast markers. The authors focused their analysis on fibroblasts, monocytes, macrophages and T cells.

The authors found that fibroblasts in IPF lungs are not only increased in numbers but also differ phenotypically from control lung fibroblasts, as demonstrated by a principal component analysis of the levels of six selected markers (PD-L1, PD-L2, CD47, CALRETICULIN, FSP1, PODOPLANIN and PDGFRA) in fibroblasts. Further analysis of the mass cytometry data showed activation of JUN and AKT in 50% of fibroblasts from human IPF lungs when compared to control lungs. In addition, more than 30% of the fibroblasts from IPF lungs expressed CD47 and a subset (~10%) coexpressed PD-L1. The author partially confirmed these results by immunostainings in lung sections and qRT-PCR-based expression analysis.

Among all CD45+ immune cells contained in the single cell suspensions analyzed by the authors, the numbers of T cells, dendritic cells and B cells were increased (~2-fold) in IPF lungs when compared to control lungs. Interestingly, while no quantitative differences in macrophage numbers were observed, macrophages from IPF lungs were phenotypically different from control lungs as demonstrated by principal component analysis. The authors also detected that most functional macrophage markers were down regulated. In addition, the alveolar to interstitial macrophage ratios were severely perturbed.

A detailed characterization of T cells demonstrated that specific subsets of naive CD4, naive CD8 and Th2 T cells were decreased in the single cell suspensions from IPF lungs when compared to control lungs. No difference between IPF and control samples was detected for Th1 and Th17 T cells. Interestingly, strong increased numbers of exhausted T cells, regulatory T cells and PD-1+CD4+ T

cells were detected in IPF samples when compared to control samples, which suggest an immunosuppressive microenvironment. The mass cytometry results were partially confirmed by immunostainings in lung sections.

The authors analyzed the chromatin from lung fibroblasts of IPF patients by sequencing after Assay for Transposase Accessible Chromatin (ATAC-seq). The authors investigated the effect of CRISPR-Cas9-induced genetic ablation of JUN on chromatin accessibility by ATAC-seq. The authors found that chromatin peaks from genes involved in pro-fibrotic epithelial-mesenchymal transition pathway (ACTA2, PDGFRB, COL1A2 and COL6A3) were significantly lost (log FDR-value 10⁻¹⁵) after JUN deletion. In addition, pro-inflammatory genes (GLI1, NFKB1 and IL6 receptor) also appeared to be regulated and downstream of JUN as their chromatin accessibilities decreased in JUN knockout (KO). In addition, the authors compared their ATAC-seq data with published gene expression profiling (GEP) from IPF and healthy lungs and found an overlap of 70 genes between these two datasets. Among these overlapping genes the most significant were genes encoding the pro-fibrotic epithelial-mesenchymal transition pathway, indicating that the JUN could be key regulator mediating lung fibrosis in IPF. In fact, the authors speculate that JUN might directly regulate CD47 and PDL-1 at the transcriptional level in the context of lung fibrosis as previously reported for MYC in the context of cancer. The authors refer to their previous work based on a transgenic mouse model and also present new data using different experimental systems, which support their hypothesis that JUN is a master regulator of lung fibrosis in IPF.

The authors used a multiplex assay to profile chemokines in bronchoalveolar lavage (BAL) from the same IPF patients that were analyzed by mass cytometry. The levels of the cytokine IL6 was relatively high in the BALs from IPF patients when compared to control donors. The authors demonstrate that the high levels of IL6 are JUN dependent in human IPF, as well as in mouse pro-fibrotic lung fibroblasts. The authors suggest that IL6 is a critical downstream cytokine pathway involved in the JUN-mediated pro-fibrotic response. Moreover, the authors demonstrate that IL6-signaling amplifies some of the pro-fibrotic effects caused by JUN.

The authors used the bleomycin mouse model for a semi-therapeutic approach, in which they tested the blockade of PD-1/PD-L1, CD47 and IL6 alone or in combination of two or three blockades simultaneously. The effect of the treatment was monitored by serial high-resolution CT imaging of the mouse lung weekly. The authors found a reduction of the fibrosis in the lung highlighted by reduced radio densities, a result most notable in lungs of mice treated with triple combined PD-L1, CD47 and IL6 blockade. The lung of these mice were also analyzed by mass cytometry and histopathological and immunostainings. Further, the effect of this treatment on the innate immunity was analyzed by phagocytosis assays. In addition, the authors established a humanized mouse model of IPF, in which they successfully engrafted primary human IPF lung fibroblasts in NOD-SCID-IL-2Rg KO mice underneath the kidney capsule to evaluate in vivo the response to treatment of human IPF and to assess the efficiency of PD-1/PD-L1 blockade on the innate immunity in the absence of T cells.

Conceptually, the characterization of the heterogeneous population of cells that is relevant for the pathophysiology of human idiopathic pulmonary fibrosis is novel and certainly one of the strengths of the manuscript. While single cell deconvolution of the fibroblast heterogeneity was recently reported in a bleomycin mouse model for pulmonary fibrosis, no comprehensive single cell data are yet available for human IPF. The data presented by the authors will be a major contribution to the IPF field and also of interest for scientists working on fibrosis of other organs. The concept of JUN being a master regulator of lung fibrosis in IPF is not new. The authors demonstrated in one of their previous publications that activation of JUN in a transgenic mouse model is sufficient to induce lung fibrosis. However, the proposed mechanism of JUN mediating changes in chromatin structure that regulates the expression of pro-fibrotic genes and immune checkpoint pathways genes is in turn novel. Other strength of the manuscript is that the authors used elaborated and of state-of-the-art methods for obtaining the data supporting their claims. The clinical relevance of the manuscript is reflected in the data that support a therapeutic approach against IPF based on a triple combined PD-L1, CD47 and IL6 blockade. Even though these are all strengths of the manuscripts, the current version of the manuscript is not suitable for publication at Nature Communications. There are various major concerns that have to be addressed in order to improve the quality of the manuscript, thereby achieving the standards for publications at Nature Communications.

Major concerns:

1. The general structure of the manuscript has to be improved. The number of main figures has to increase from four to at least seven main figures, for example by moving data from the supplementary figures to the main figures. The current version of the main figures and supplementary figures is not comprehensive. The reader cannot recognize in the figures what is described in the Results section. The size of various panels has to be increased and the labelling of the figures has to be improved. Moreover, the description of the data in the results section has to be more accurate, for example describing the statistical relevance of the data presented. The authors should take care of a correct use of terms, what is not always the case in the present version of the manuscript. The text is repetitive in various parts of the manuscript. The authors should take care of using the right references. By improving all these aspects, the scientific value of the presented data will be easier to recognize.

2. The data suggesting the mechanism of JUN as master regulator of lung fibrosis in IPF mediating changes in chromatin structure that regulates the expression of pro-fibrotic genes and immune checkpoint pathways are not robust enough to support the interpretation of the authors. Additional data have to be presented supporting the claims of the author.

- 2.1 The authors claim that JUN controls the expression of CD47, PDL-1 and IL6 mainly based on results obtained after loss-of-function (LOF) experiments in IPF fibroblasts. Since the authors have identified different sub populations of IPF fibroblasts, which of them have been used by the authors

in their analysis? Furthermore, to strengthen the interpretation of their LOF experiments, the authors should perform gain-of-function (GOF) experiments, in which they activate JUN in healthy fibroblasts and evaluate the effects on chromatin structure and expression levels of CD47, PDL-1 and IL6. In addition to these single gene analyses, the authors should consider analyzing the chromatin in control fibroblasts without or with activation of JUN by a ChIP-seq experiment using JUN specific antibodies. The combinatorial analysis of the ChIP-seq experiment with the presented ATAC-seq experiment should confirm the key role of JUN in lung fibrosis during IPF, thereby confirming the claims of the authors.

2.2 Following this line of ideas, in a similar experimental setting as described in the previous paragraph, the author might consider to use in the ChIP-seq experiment antibodies specific for open chromatin (H3K4me3, H3K27Ac), close chromatin (K3K9me3, H3K27me3) and for active RNA polymerase II. The data obtained in this ChIP-seq experiment should reveal the genome wide chromatin landscape and the transcriptional stage in the cells analyzed.

2.3 The authors claim that CD47, PDL-1 and IL6 are epigenetically regulated by JUN. However, there is no analysis regarding DNA methylation, histone modifications, histone variant deposition, etc. Chromatin accessibility is the consequence of changes in the chromatin structure. Epigenetic regulation implies changes in the chromatin structure that are transmitted to the “daughter cells” after cell division. The term “epigenetic” should be used properly.

2.4 The authors found in their ATAC-seq that JUN-LOF affects CD47 promoter and enhancer. However, in their single gene analysis to confirm the ATAC-seq results the author focused only on the enhancer. Why the authors did not analyze the effect on the CD47 promoter? Why the authors do not include PDL-1 and IL6 in their single gene analysis confirming the ATAC-seq results?

2.5 Overexpression of the CD47 enhancer construct for 6 days might force the system to its limit. To better support their findings, the authors should perform ChIP experiments for JUN at the target regions (as described under 2.1).

2.6 What is represented in the snap shots in Figure S4D? The authors should specify each line and scales. The authors should validate that JUN binds to IL6 and that this is not a secondary effect (see 2.1). Why do the authors identify major changes only downstream the TSS and not in the promoter?

2.7 The authors activate JUN-mediated CD47 enhancer activity by IL6 treatment. Are the concentrations physiological relevant? How the treatment does affect the viability and the growth of the cells?

2.8 The results presented in the Figure S1D are not conclusive. The authors have to explain better the heterodimer formation between FOS and JUN mentioned in Figure S1D.

3. The authors did not provide access to the ATAC-seq. Thus, I was not able to confirm the quality of the data, neither the data analysis. The same issue is related to the published RNA-seq and ChIP-seq that the author used in the manuscript to compare with their ATAC-seq data. The author has to provide the GEO accession number in order that the reader can clearly identify the data set used, in case that the reader would like to reproduce the analysis described by the authors.

4. In the immunofluorescence presented in Figure 1H, the authors attempt to confirm the results obtained by mass cytometry. However, the authors did not present the results from co-immunostaining using antibodies specific for CD47 and PD-L1 and demonstrating the presence of these two proteins in 10% of lung fibroblasts. According to panel 1G they should detect around 10% of cells in which both proteins are present. In addition, the authors forgot to place the scale bars in panel 1H.

5. For the principal component analysis of lung fibroblasts after mass cytometry, the authors selected six markers: PD-L1, PD-L2, CD47, CALRETICULIN, FSP1, PODOPLANIN and PDGFRA. The author should explain in the text the criteria of selection of these markers, and whether other fibroblasts related markers were not included in this principal component analysis.

6. The authors analyzed single cell suspensions of 14 representative lung samples (11 IPF samples and 3 control samples). The authors have to provide clinical data related to the samples used. For example: IPF stage, gender, age, smoking history, other pathologies, etc. If the IPF samples were from different disease stages, this might explain the increase on the fibroblast percentage between 50 to 90% observed in the in figure 1C.

7. Is the observation from Figure 1I, claiming an increase of PD-L1 and CD47 expression in IPF when compared to control lungs, sustained by the expression profile in RNA-seq experiments published and used by the authors to confirm the ATAC-seq (see reference 35 and figure S3B)? As mentioned under point 3, the GEO accession number of the RNA-seq data has to be provided in the manuscript.

8. The authors conclude that suppressive immune cell types predominate in the IPF lung, but they might have preferential topology along the lung tissue. What would be the tissue infiltration rate from different pulmonary sections?

9. Several functional assays in primary fibroblasts (proliferation, migratory profile, Sircoll, Sirius Red, hydroxyproline, etc.) are required upon inhibition of CD47, PD-L1 and IL6.

10. The analysis demonstrating the statistical relevance of the data should be presented in the manuscript. P-values and the test used for the calculation should be presented in the figures and figure legends. In addition, the author might consider to summarize all these information related to all the data presented in the manuscript in a supplementary excel file.

Minor concerns:

11. Scale bars in all microscopy pictures should be presented.

12. Labelling of the figure panels is not uniform with respect of the lettering size. In addition, figure panels are too small to recognize the results

13. References should be mentioned correctly. For example on page 3 "...While single cell deconvolution of the fibroblast heterogeneity was recently reported in a bleomycin initiated pulmonary fibrosis mouse model, no comprehensive single cell protein data focusing on fibroblast heterogeneity are yet available for human IPF (17)." The correct reference for this statement is the reference number 18.

14. On page 4 the authors wrote: "In addition to the increased abundance of fibroblasts, we performed a principal component analysis of the expression level of six markers (PD-L1, PD-L2, CD47, CALRETICULIN, FSP1, PODOPLANIN and PDGFRA) on fibroblasts and demonstrated that IPF lung fibroblasts from 13 IPF patients clustered together and were distinct from lung fibroblasts derived from normal healthy lungs (Figure 1D),...". However, in the figure and in other part of the text is written that 11 IPF patients were analyzed.

15. The author should introduce abbreviations at the first place used, and implement them consequently along the whole manuscript.

16. The author should avoid repetition of the text thereby reducing redundancy.

The data presented in the manuscript will be a major contribution for the field of IPF and of interest for scientists working on fibrosis in other organs. In addition, the manuscript has a strong translational potential suggesting therapeutic approaches against IPF. I strongly believe that after addressing all the concerns, the manuscript will improve significantly achieving the standards for publication at Nature Communications.

Reviewer #2 (Remarks to the Author):

The manuscript submitted by Cui and colleagues entitled “Activation of JUN in fibroblasts promotes pro-fibrotic programme and dampens protective immunity” utilizes CyTOF to assess a panel of markers in immune cells and fibroblasts isolated from lung tissue of 11 humans with idiopathic pulmonary fibrosis (IPF) and 3 “healthy” controls. The authors report differences in the relative frequency of leukocytes and mesenchymal cells between the two groups and show that fibroblasts from the IPF subjects have increased expression of CD47, PDL1 and c-JUN. They also report increased expression of PD1 in macrophages and lymphocytes isolated from the IPF subjects. Using CRIPR-Cas9, the investigators delete JUN in IPF fibroblasts and show that this regulates CD47 enhancer activity and downstream expression of CD47. While the enhancer landscape of PDL1 was altered by JUN deletion in fibroblasts, downstream expression of PDL1 was not shown. Addition of IL-6 to fibroblast cultures increased CD47 expression. In an attempt to link PDL1, CD47 and IL-6 to fibrosis, the investigators blocked these pathways using antibodies, inhibitors and IL-6 knockout mice. Various combinations led to attenuated fibrosis. Lastly, the investigators employed a model in which human fibroblasts (IPF and normal) were transplanted into the kidney capsule of immunodeficient mice (Rag2ko mice lacking IL2R). Inhibition of PD1/PDL1 reduced fibroblast bioluminescence in the IPF group and attenuated the degree of fibrosis.

The experiments presented in the manuscript employ state-of-the-art technologies and leverage human tissue samples for mechanistic studies. Clearly, a huge amount of work went into these studies. While the role of JUN as a driver of fibrosis is well recognized, the identification of JUN as a regulator of CD47 (and possibly PDL1 – although not fully shown) is novel. However there are a number of concerns with the manuscript as detailed below.

Major concerns.

1. The manuscript is difficult to follow and many key details are lacking (see below). It is difficult to follow many of the experiments and to know exactly how they were performed. Critical controls are lacking for some experiments. This leaves the reader with the impression that there was insufficient attention to detail.
2. The manuscript has a number of pieces of data that aren't completely linked or explored. This leaves the manuscript somewhat scattered, and contributes to the overall difficulty with its interpretation. Moreover, many of the conclusions of the manuscript are overstated and not supported by sufficient experimental exploration. Just one example is CD47. The investigators provide solid evidence that fibroblasts from IPF patients have increased levels of CD47 and that JUN promotes CD47 expression. However, how CD47 contributes to fibrosis is not explored. For instance, would blockade of CD47 by itself dampen the fibrotic response? If so, how would this effect occur? Is expression of CD47 ligands (such as SIRP alpha) increased on other cells? One way to rescue the manuscript would be to trim it down and to tell a single concise story that is well supported by a full complement of experiments.

3. The sources of tissue that comprise the IPF versus “healthy” samples are markedly different. This is likely to influence results.

a. The IPF tissue comes from explanted lungs whereas the “healthy” tissue comes from surgical resections (presumably this tissue is peri-tumoral). In the Methods section, the authors state that they tried to obtain post-mortem tissue but that viability was poor. While I empathize with the major challenges presented in obtaining “healthy” tissue, I am concerned that at least some of the results in Figs 1 and 2 may arise from this. (For instance, a marked absence in endothelial cells in “healthy” tissue and an absence of epithelial cells in IPF tissue).

b. Is it possible that some of the differences noted between the healthy and IPF tissues arises from the fact that the IPF tissue could contain larger airways and vessels whereas the “healthy” tissue will primarily contain alveolar regions. Fibroblasts, and other cell types may vary based on this fact alone.

c. The IPF versus “healthy” tissues should be more completely described in the methods. Of particular importance is knowing whether the “healthy” tissue samples were cryopreserved and whether the IPF samples were fresh (or also cryopreserved). The ages, sex, smoking status, etc should be given in a table (perhaps in the supplement).

d. BAL fluid from IPF and healthy controls are used. How were the BAL performed? Was the same volume instilled between groups and were returns comparable? Was the same technique used in both cases? Was the BAL from IPF subjects performed on post-mortem lung?

4. The manuscript title suggests that JUN expression in fibroblasts dampens protective immunity. This is not shown experimentally.

5. The ATAC seq studies suggest that PDL1 is downstream of JUN. This is not confirmed experimentally (as was nicely done with CD47). Along similar lines, the investigators suggest that IL-6 may enhance JUN expression / activity and that this will increase PDL1. Experimental evidence is lacking.

6. Tissue immunohistochemistry results are not convincing. First, negative controls are not shown and it is possible that at least some of the signal arises from autofluorescence. Second, stains to

show tissue architecture are lacking. H&E stains from serial sections (or at a minimum DIC images) would help orient the reader. Third, it is unclear whether the sections from the IPF subjects were taken at random or from areas with the greatest fibrosis. Since sections from 11 subjects were analyzed, some effort at quantification of the results should be performed.

Additional comments.

The tissues from the IPF subjects appear to have a paucity of alveolar macrophages (AM) whereas the healthy control tissues have few, if any, interstitial macrophages (IM). Accordingly, the studies that compare macrophages from the IPF versus control may be influenced by markedly different ratios of the IM and AMs. An advantage to CyTOF is that the different cell types can be selected and compared head-to-head.

The term “blood cells” is used throughout the manuscript and is quite confusing. The term “leukocyte is preferable”. Many of the cells that are isolated are likely to be outside of the circulation.

Gating strategies for the different cell types should be shown in the CyTOF experiments. This is especially true for macrophages and for fibroblasts. In terms of the latter, the investigators define fibroblasts as CD45- CD31- CK7- cells. It should be noted that this scheme would include other mesenchymal cells such as smooth muscle cells.

The text confuses total cell numbers with the frequency of cells in several locations. I do not see that total cell numbers were ever evaluated for the tissue. An example of this occurs in the first paragraph of the Results section in which the authors state “we detected a 5-fold increase in fibroblasts...” It would be better to say that the frequency of fibroblasts was 5-fold higher in IPF tissues. Later in the same paragraph the text states “... fibroblasts in IPF lungs are not only increased in numbers...” While there are likely to be more fibroblasts in the IPF tissues, the data do not show this. They only show an increased percentage.

Figure 1G. Presumably these are flow cytometry data (although not explicitly stated). It would be informative to show gating strategies used to arrive at the putative fibroblasts in the Supplement. In this regard, a panel with live-dead discrimination should be shown as well as FMO controls to ensure that changes in autofluorescence are not driving the putative differences.

Figure 1J shows increased PDL1 protein in BAL from IPF lungs. What is the point of showing this? Is it assumed that PDL1 from the airspaces can gain access to the interstitium and have an effect on the fibroblasts?

Figure 2. The legend does not completely match the figure. This makes the figure very hard to interpret. In Figure 2A nearly 40% of cells are “other.” What are these cells? Can their identity be determined? If not, this brings the data from the rest of Figure 2A into question. Figure 2B doesn’t add much to the manuscript. Figure 2C and 2D are confusing. It is not clear what populations I and II are, nor is it clear in Figure 2D whether cells are from healthy lungs, IPF lungs, or both.

Figure 3. It is curious that CD47 does not show up in Fig 3A. How do the authors explain this? Figs 3D and E demonstrate that Jun is upstream of CD47 in cultured fibroblasts. Within these experiments doxycycline is used to induce Jun expression. This effect of doxy is not mentioned anywhere in the text, and unless the reader is aware of the group’s previous work, these experiments are impossible to interpret. The text should be expanded to explain this.

Figure S4 shows IL-6 expression in a number of different cell types in culture. In these experiments c-JUN expression was reportedly induced with doxycycline. Is it possible that the increases in IL-6 are a result of off-target effects? It would be important to show that JUN was induced, and that inhibition of c-JUN (perhaps via an inhibitor) blocks IL-6.

Kidney capsule studies with transplanted fibroblasts are intriguing, however more intense analysis would be reassuring. First, it appears that luminescence from the HAC- control fibroblasts is low at baseline. This is confusing, and suggests that there is a problem – perhaps low numbers of healthy fibroblasts were transplanted? Second, it is implied that fibroblasts from the HAC group are being removed by macrophages. Stains to show macrophages in the tissue would be helpful to support this – particularly if they show engulfment of fibroblasts. Finally, it would be instructive to know if the IPF fibroblasts are dying. This could be achieved by staining the tissues.

Reviewer #4 (Remarks to the Author):

The manuscript from Cui et al. clearly describes a mechanism for the evolution of a fibrotic program in the lung. Experiments try to logically dissect the mechanism and overall gave strong evidence that JUN activation is key for the pro-fibrotic program. The use of data generated in humans and the final demonstration of the mechanism in a mouse model strongly support the hypothesis.

Could the authors make available the CyTOF FCS files?

In Figure 2 A and B the title "Blood Cells" seems to indicate that the cells are derived from blood whereas these are cell collected from the lungs

Figure 2C show a tSNE analysis whereas in the text (page 6, line7) and figure legend is referred to as PCA. The legend of Figure 2F state that the Figure shows IDO expression whereas it shows the PCA. The Authors need to revise the entire legend and the references in the text.

The separation of single cells in the tSNE plots in Figure 2D is really poor. Will be possible to run the algorithm with more iterations or change the perplexity? I have the same observation for some of the other tSNE plots throughout the manuscript

It will be interesting to show in Supplementary Figure 2 the expression of CD47

Could the Authors add a reference for the involvement of PDGF-BB and CCL5 in connective tissue remodeling (page 8)?

Why the Authors finally use the Bleomycin mouse model instead of the JUN-inducible mouse model for the checkpoint blockade experiments?

Could the author build a Supplementary table summarizing the number of living cells found in each sample and the number of cells used to run each algorithm?

Supplementary Table 4: please indicate the fluorochromes for each antibody

In the Methods section, MIBI technology is not described

Nature communications reviewer comments:

Reviewer #1

Peer review of the manuscript with the number NCOMMS-19-12042 at Nature Communications by Cui L, Chen SY et al and with the title: "Activation of JUN in fibroblasts promotes pro-fibrotic programme and dampens protective immunity"

The authors characterized the heterogeneous population of cells that is relevant for the pathophysiology of human idiopathic pulmonary fibrosis. The authors analyzed single cell suspensions of 14 representative lung samples (11 IPF samples and 3 control samples) by mass cytometry after staining with a panel of 41 metal conjugated antibodies. The author selected the antibodies used for the mass cytometry analysis based on published literature. The authors claim to have selected antibodies specific against the most published canonical fibroblast markers. The authors focused their analysis on fibroblasts, monocytes, macrophages and T cells.

The authors found that fibroblasts in IPF lungs are not only increased in numbers but also differ phenotypically from control lung fibroblasts, as demonstrated by a principal component analysis of the levels of six selected markers (PD-L1, PD-L2, CD47, CALRETICULIN, FSP1, PODOPLANIN and PDGFRA) in fibroblasts. Further analysis of the mass cytometry data showed activation of JUN and AKT in 50% of fibroblasts from human IPF lungs when compared to control lungs. In addition, more than 30% of the fibroblasts from IPF lungs expressed CD47 and a subset (~10%) coexpressed PD-L1. The author partially confirmed these results by immunostainings in lung sections and qRT-PCR-based expression analysis.

Among all CD45+ immune cells contained in the single cell suspensions analyzed by the authors, the numbers of T cells, dendritic cells and B cells were increased (~2-fold) in IPF lungs when compared to control lungs. Interestingly, while no quantitative differences in macrophage numbers were observed, macrophages from IPF lungs were phenotypically different from control lungs as demonstrated by principal component analysis. The authors also detected that most functional macrophage markers were

down regulated. In addition, the alveolar to interstitial macrophage ratios were severely perturbed.

A detailed characterization of T cells demonstrated that specific subsets of naive CD4, naive CD8 and Th2 T cells were decreased in the single cell suspensions from IPF lungs when compared to control lungs. No difference between IPF and control samples was detected for Th1 and Th17 T cells. Interestingly, strong increased numbers of exhausted T cells, regulatory T cells and PD-1+CD4+ T cells were detected in IPF samples when compared to control samples, which suggest an immunosuppressive microenvironment. The mass cytometry results were partially confirmed by immunostainings in lung sections.

The authors analyzed the chromatin from lung fibroblasts of IPF patients by sequencing after Assay for Transposase Accessible Chromatin (ATAC-seq). The authors investigated the effect of CRISPR-Cas9-induced genetic ablation of JUN on chromatin accessibility by ATAC-seq. The authors found that chromatin peaks from genes involved in pro-fibrotic epithelial-mesenchymal transition pathway (ACTA2, PDGFRB, COL1A2 and COL6A3) were significantly lost (log FDR-value 10⁻¹⁵) after JUN deletion. In addition, pro-inflammatory genes (GLI1, NFKB1 and IL6 receptor) also appeared to be regulated and downstream of JUN as their chromatin accessibilities decreased in JUN knockout (KO). In addition, the authors compared their ATAC-seq data with published gene expression profiling (GEP) from IPF and healthy lungs and found an overlap of 70 genes between these two datasets. Among these overlapping genes the most significant were genes encoding the pro-fibrotic epithelial-mesenchymal transition pathway, indicating that the JUN could be key regulator mediating lung fibrosis in IPF. In fact, the authors speculate that JUN might directly regulate CD47 and PDL-1 at the transcriptional level in the context of lung fibrosis as previously reported for MYC in the context of cancer. The authors refer to their previous work based on a transgenic mouse model and also present new data using different experimental systems, which support their hypothesis that JUN is a master regulator of lung fibrosis in IPF.

The authors used a multiplex assay to profile chemokines in bronchoalveolar lavage (BAL) from the same IPF patients that were analyzed by mass cytometry. The levels of the cytokine IL6 was relatively high in the BALs from IPF patients when compared to

control donors. The authors demonstrate that the high levels of IL6 are JUN dependent in human IPF, as well as in mouse pro-fibrotic lung fibroblasts. The authors suggest that IL6 is a critical downstream cytokine pathway involved in the JUN-mediated pro-fibrotic response. Moreover, the authors demonstrate that IL6-signaling amplifies some of the pro-fibrotic effects caused by JUN.

The authors used the bleomycin mouse model for a semi-therapeutic approach, in which they tested the blockade of PD-1/PD-L1, CD47 and IL6 alone or in combination of two or three blockades simultaneously. The effect of the treatment was monitored by serial high-resolution CT imaging of the mouse lung weekly. The authors found a reduction of the fibrosis in the lung highlighted by reduced radio densities, a result most notable in lungs of mice treated with triple combined PD-L1, CD47 and IL6 blockade. The lung of these mice were also analyzed by mass cytometry and histopathological and immunostainings. Further, the effect of this treatment on the innate immunity was analyzed by phagocytosis assays. In addition, the authors established a humanized mouse model of IPF, in which they successfully engrafted primary human IPF lung fibroblasts in NOD-SCID-IL-2Rg KO mice underneath the kidney capsule to evaluate in vivo the response to treatment of human IPF and to assess the efficiency of PD-1/PD-L1 blockade on the innate immunity in the absence of T cells.

Conceptually, the characterization of the heterogeneous population of cells that is relevant for the pathophysiology of human idiopathic pulmonary fibrosis is novel and certainly one of the strengths of the manuscript. While single cell deconvolution of the fibroblast heterogeneity was recently reported in a bleomycin mouse model for pulmonary fibrosis, no comprehensive single cell data are yet available for human IPF. The data presented by the authors will be a major contribution to the IPF field and also of interest for scientists working on fibrosis of other organs. The concept of JUN being a master regulator of lung fibrosis in IPF is not new. The authors demonstrated in one of their previous publications that activation of JUN in a transgenic mouse model is sufficient to induce lung fibrosis. However, the proposed mechanism of JUN mediating changes in chromatin structure that regulates the expression of pro-fibrotic genes and immune checkpoint pathways genes is in turn novel. Other strength of the manuscript is that the authors used elaborated and of state-of-the-art methods for obtaining the data

supporting their claims. The clinical relevance of the manuscript is reflected in the data that support a therapeutic approach against IPF based on a triple combined PD-L1, CD47 and IL6 blockade. Even though these are all strengths of the manuscripts, the current version of the manuscript is not suitable for publication at Nature Communications. There are various major concerns that have to be addressed in order to improve the quality of the manuscript, thereby achieving the standards for publications at Nature Communications.

Major concerns:

1. The general structure of the manuscript has to be improved. The number of main figures has to increase from four to at least seven main figures, for example by moving data from the supplementary figures to the main figures. The current version of the main figures and supplementary figures is not comprehensive. The reader cannot recognize in the figures what is described in the Results section. The size of various panels has to be increased and the labelling of the figures has to be improved. Moreover, the description of the data in the results section has to be more accurate, for example describing the statistical relevance of the data presented. The authors should take care of a correct use of terms, what is not always the case in the present version of the manuscript. The text is repetitive in various parts of the manuscript. The authors should take care of using the right references. By improving all these aspects, the scientific value of the presented data will be easier to recognize.

We thank reviewer #1 for this comment and have revised the figures and manuscript accordingly. We now include supplemental table 4 and 5 describing the statistical relevance in details and the definition of terms.

2. The data suggesting the mechanism of JUN as master regulator of lung fibrosis in IPF mediating changes in chromatin structure that regulates the expression of pro-fibrotic genes and immune checkpoint pathways are not robust enough to support the

interpretation of the authors. Additional data have to be presented supporting the claims of the author.

We agree with the reviewer, this is an important point. While we did not directly refer to *JUN* as a master regulator, but a critical regulator in lung fibrosis, previous work by Vierbuchen et al. (reviewed by Madrigal and Alasoo, 2018)^{1,2} demonstrates that *JUN*, as part of the AP-1 (*FOS/JUN*) complex, can function as a pioneer transcription factor and acts as an enhancer selector to modulate the accessibility of DNA in fibroblasts. Thus, *JUN* is one of the key factors that can remodel chromatin, increase DNA accessibility to regulate the expression of genes of fibrosis.

To support our interpretation regarding *JUN* as a regulator of the expression of pro-fibrotic genes (*IL6*) and immune checkpoint pathways (*CD47*, *PDL1*(*CD274*)), we followed the reviewer's advice (thank you for the very helpful comment) and performed ATAC-seq and *JUN* ChIP-seq on primary lung fibroblasts from normal lung samples with or without *JUN* overexpression (see **Fig.4**), in addition to the ATAC-seq on fibrotic lung fibroblasts we previously performed with or without *JUN* knockout. First the new *JUN* ChIP-seq data coupled with the ATAC data confirms enrichment of bound JUN to the *JUN* promoter region (shaded in red), which correlates with accessible chromatin state (detected by ATAC-seq) in overexpressed *JUN* lung fibroblasts when compared to normal cells. This demonstrates that overexpression of *JUN* increases accessibility to its own promoter (in a positive regulatory feedback fashion). For the pro-fibrotic gene *IL6*, we found similar results: JUN binds to the promoter (shaded in red) of *IL6* in normal cells and its binding is highly enriched in the overexpressed JUN lung fibroblast cells, thus, increasing *IL6* promoter accessibility in these cells (detected by ATAC-seq)

However, when we analyzed the DNA bound-JUN effects on chromatin structure of *CD47* or *PDL1*, we noticed that *JUN* enrichment (by *JUN* ChIP-seq) occurs preferentially in a distal genomic region (shaded in green) for *CD47* and in an intronic genomic region (shaded in green) for *PDL1*, rather than in their corresponding promoters (shaded in red) the *JUN* enrichment observed in these two cases correlated

with an increase in chromatin accessibility (detected by ATAC-seq) in lung fibroblast cells when compared to normal. This is particularly interesting as these changes are only present in our primary lung fibroblasts but not in any of the other previously published data on *JUN* ChIP-seq performed on cancer cell lines such as A549, MCF7, H1-hESC, HepG2 or K562, the links are included below. Our new results suggest that the binding of JUN to specific *CD47* and *PDL1* regions might modulate accessibility to DNA in regulatory regions specific to fibrotic disease. Such interesting result will be followed up in future work. We thank the reviewer for also questioning why our first ATAC-seq analyses focused primarily on promoter regions. Now the data showing JUN binding to enhancers and effects is included.

JUN ChIP-seq on human HepG2

GEO:GSM935364

<https://www.encodeproject.org/experiments/ENCSR000EEK/>

JUN ChIP-seq on human MCF-7

GEO:GSE91550

<https://www.encodeproject.org/experiments/ENCSR176EXN/>

JUN ChIP-seq on human H1-hESC

GEO:GSM935614

<https://www.encodeproject.org/experiments/ENCSR000ECA/>

JUN ChIP-seq on human A549

GEO:GSE92221

<https://www.encodeproject.org/experiments/ENCSR996DUT/>

JUN ChIP-seq on human K562

GEO:GSM1003609

<https://www.encodeproject.org/experiments/ENCSR000EFS/>

2.1 The authors claim that JUN controls the expression of CD47, PDL-1 and IL6 mainly based on results obtained after loss-of-function (LOF) experiments in IPF fibroblasts. Since the authors have identified different sub populations of IPF fibroblasts, which of them have been used by the authors in their analysis?

Thank you for your comment. Here we show flow cytometry data of human pulmonary fibrosis lung fibroblasts in steady state as well as q-PCR data and find an excellent

correlation of JUN and CD47 and PDL1 expression. We characterized our fibroblast population by using the following marker: CD45- (PE-Cy7), CD326- (APC), CD31- (Alexa 488), JUN+, pJUN+, CD47+ PDGFRa- PDGFRb-.

Furthermore, to strengthen the interpretation of their LOF experiments, the authors should perform gain-of-function (GOF) experiments, in which they activate JUN in healthy fibroblasts and evaluate the effects on chromatin structure and expression levels of CD47, PDL-1 and IL6.

In **Fig. 4f, g** (in the revised manuscript) we evaluated the effects of JUN-GOF and JUN-LOF in parallel. We show increased reporter activity for the *CD47* enhancer with DOX induced JUN overexpression and decreased reporter activity for the *CD47* enhancer with DOX withdrawal (JUN off) or JUN knock out.

In addition, in **Fig. 4c, d** we assessed protein expression by flow and RNA expression by qPCR for JUN-GOF (see data below). We show that the expression levels of IL6, PD-L1 and CD47 are increased with JUN overexpression.

In addition to these single gene analyses, the authors should consider analyzing the chromatin in control fibroblasts without or with activation of JUN by a ChIP-seq experiment using JUN specific antibodies.

We thank the reviewer for this excellent suggestion. As suggested by the reviewer, we have performed ChIP-seq with JUN specific antibodies on wildtype lung fibroblasts without and with *JUN* activation and we describe these data below and in **Fig. 4** of the revised manuscript and under comments 2.2-2.6.

In addition, we have performed ATAC seq experiments on wildtype lung fibroblasts without and with *JUN* overexpression and found 38,240 peaks which are differentially responding to *JUN* expression such as CD47, PDL1, PDL2, collagens.

The combinatorial analysis of the ChIP-seq experiment with the presented ATAC-seq experiment should confirm the key role of JUN in lung fibrosis during IPF, thereby confirming the claims of the authors.

Thank you for this highly valuable suggestion, we performed the ChIP-seq experiments as recommended by this reviewer and discuss the findings in **Fig. 4** of the revised manuscript and under comments below 2.2-2.6.

2.2 Following this line of ideas, in a similar experimental setting as described in the previous paragraph, the author might consider using in the ChIP-seq experiment antibodies specific for open chromatin (H3K4me3, H3K27Ac), close chromatin (K3K9me3, H3K27me3) and for active RNA polymerase II. The data obtained in this ChIP-seq experiment should reveal the genome wide chromatin landscape and the transcriptional stage in the cells analyzed.

We thank the reviewer for this valid suggestion. For the experiments performed in this study we used precious primary patient samples, from which the total number of viable cells is very limited. We chose to perform ATAC-seq because it is a technique that can be performed on relatively low cell numbers, and it can still detect chromatin accessibility changes accurately (which was the goal of this experiment). Although we agree with the reviewer that performing ChIP-seq experiment using antibodies specific for open chromatin markers, close chromatin markers and for active RNA polymerase II, will reveal in a very detailed manner the genome wide chromatin landscape for all genes and the transcriptional stage of the cells analyzed. This is outside of the scope of this study. We unfortunately do not have the amount of starting material required for doing this type of profiling and the material we had, we used it to address the translational aspects of this study, which were our main focus. However, in an effort to

address the reviewer's important suggestion, we now include H3K27me3 ChIP-seq data in addition to H3K27Ac tracks which were already included previously (links are listed below). This data together with our previous data supports that DNA accessibility driven by JUN dramatically changes in response to the presence or absence of JUN.

H3K4me3 ChIP-seq on human NHLF*

GEO:GSM733723

<https://www.encodeproject.org/experiments/ENCSR000AMW/>

H3K27ac ChIP-seq on human NHLF*

GEO:GSM733646

<https://www.encodeproject.org/experiments/ENCSR000AMR/>

H3K9me3 ChIP-seq on human NHLF*

GEO:GSM1003531

<https://www.encodeproject.org/experiments/ENCSR000ARQ/>

H3K27me3 ChIP-seq on human NHLF*

GEO:GSM733764

<https://www.encodeproject.org/experiments/ENCSR000AMS/>

*NHLF: normal human lung fibroblasts

2.3 The authors claim that CD47, PDL-1 and IL6 are epigenetically regulated by JUN. However, there is no analysis regarding DNA methylation, histone modifications, histone variant deposition, etc. Chromatin accessibility is the consequence of changes in the chromatin structure. Epigenetic regulation implies changes in the chromatin structure that are transmitted to the “daughter cells” after cell division. The term “epigenetic” should be used properly.

We thank the reviewer for this valid comment and we apologize for using the incorrect terminology. We deleted "epigenetic regulation" (previously used term) and corrected to "DNA accessibility" on pages 7-9.

2.4 The authors found in their ATAC-seq that JUN-LOF affects CD47 promoter and enhancer. However, in their single gene analysis to confirm the ATAC-seq results the author focused only on the enhancer. Why the authors did not analyze the effect on the CD47 promoter? Why the authors do not include PDL-1 and IL6 in their single gene analysis confirming the ATAC-seq results?

We thank the reviewer for this very important comment. Our new ATAC and ChIP-seq data allowed us to address this question now. In our original ATAC analysis on pulmonary fibrosis fibroblasts with or without *JUN* knockout (JUN-LOF), we found that JUN-LOF decreases DNA accessibility (detected by ATAC) to the *CD47*, *PDL1(CD274)*, and *IL6* promoters (region shaded in red, **Fig. 4b** and **Supplementary Fig. S5a**), as well as accessibility to the *CD47* enhancer (region shaded in green). Our new ATAC and *JUN* ChIP-seq data on normal human primary lung fibroblasts with or without *JUN* overexpression (JUN-GOF) confirms that JUN binds to or near to the *IL6* promoter and increases promoter accessibility (detected by ATAC-seq) when JUN is overexpressed (JUN-GOF), which in turn increases *IL6* expression at the RNA and protein level (**Fig. 5d** of the revised manuscript).

Importantly, our new ATAC and *JUN* ChIP-seq data on normal human primary lung fibroblasts with or without *JUN* overexpression (JUN-GOF) demonstrate that changes in chromatin accessibility are more pronounced in the enhancer region rather than the promoter region of *CD47*. The *JUN*-ChIP data shows that JUN binds preferentially to the *CD47* distal enhancer rather than the promoter and its binding to this region increases when *JUN* is overexpressed. Similar results, we observed for *PDL1*: JUN preferentially bind to the intronic enhancer region of *PDL1* and its binding to this region increases when JUN is overexpressed, Coelho MA et al. also report that *PDL1* first intronic region corresponds to a *PDL1* active enhancer.³

2.5 Overexpression of the CD47 enhancer construct for 6 days might force the system to its limit. To better support their findings, the authors should perform ChIP experiments for JUN at the target regions (as described under 2.1).

We thank the reviewer for this comment. Upon the reviewers request we performed *JUN* ChIP-seq and analyzed its binding to the *CD47* genomic locus (please see above under 2.4). *JUN* ChIP-seq also confirmed that JUN binds to the E7 *CD47* enhancer used in our study. In addition, we included a GFP expression timeline analysis of the TK control construct (without enhancer) to show that the EGFP reporter activity does not significantly change without the overexpression of JUN (minimal green fluorescence on day6 interpreted as background). Thus, increased GFP expression observed upon treatment in our experiments, is not due to GFP accumulation and forcing the system to its limits.

2.6 What is represented in the snap shots in Figure S4D? The authors should specify each line and scales. The authors should validate that JUN binds to IL6 and that this is

not a secondary effect (see 2.1). Why do the authors identify major changes only downstream the TSS and not in the promoter?

We thank the reviewer for this comment and apologize for not presenting the data clearly enough. We now specify each scale accordingly. We also labelled the promoters for IL6, IL6R and IL6ST more clearly, and removed neighboring genes for clarity. We also confirmed by JUN ChIP-seq that JUN binds to the IL6 promoter (please see figure below).

And previous paper has characterized most elements of the IL-6 promoter that lie within the 300 bp proximal to the start site (+1) of transcription where multiple AP-1-binding sites were found ⁴.

2.7 The authors activate JUN-mediated CD47 enhancer activity by IL6 treatment. Are the concentrations physiological relevant? How the treatment does affect the viability and the growth of the cells?

Thanks for this comment, we think the IL6 concentrations we used (1, 10, 100 ng/mL for stimulation) are physiological relevant based on reported literature (also see figure below)⁵ and our own quantitative IL6 measurements in pulmonary fibrosis patients. For example, the baseline levels of IL-6 in healthy men are around 2-10 pg/mL. Our ELISA results demonstrate IL-6 concentrations around 200 pg/mL in BAL of pulmonary fibrosis patients.

Our ELISA show the IL6 concentration of pulmonary fibrosis BAL is 200 pg/ml on average.

As we show here and in **Fig. 5e, f** of the revised manuscript, our in vitro reporter assay shows that already at the lowest IL6 concentration of 1 ng/mL the enhancer activity of CD47 already increases after 24h, however this is not yet reflected in an increased expression of the CD47 protein.

At the same time, we also assessed cell viability in response to IL6 treatment by flow and DAPI stains and did not observe any difference after IL6 treatments for the given time window.

2.8 The results presented in the Figure S1D are not conclusive. The authors have to explain better the heterodimer formation between FOS and JUN mentioned in Figure S1D.

We thank the reviewer for this comment and apologize for not providing sufficient explanation and background. We deleted "heterodimer formation" and rephrased to "we observe co-localization of *JUN* and *FOS* in fibrotic fibroblasts in the lung, which suggest that these two AP1 family members cooperate. This notion is supported by a vast amount of literature which demonstrated that *JUN* either forms a homodimer with itself or heterodimers with other AP1 family members.^{1,6}

3. The authors did not provide access to the ATAC-seq. Thus, I was not able to confirm the quality of the data, neither the data analysis. The same issue is related to the published RNA-seq and ChIP-seq that the author used in the manuscript to compare with their ATAC-seq data. The author has to provide the GEO accession number in order that the reader can clearly identify the data set used, in case that the reader would like to reproduce the analysis described by the authors.

We apologize for this oversight, the data are available on GEO, please find the link included here:

ATAC-seq and ChIP-seq on primary fibrotic and normal lung fibroblasts.

GSE115235

<https://www.ncbi.nlm.nih.gov/geo/query/acc.cgi?acc=GSE115235>

JUN ChIP-seq on human HepG2

GEO:GSM935364

<https://www.encodeproject.org/experiments/ENCSR000EEK/>

JUN ChIP-seq on human MCF-7

GEO:GSE91550

<https://www.encodeproject.org/experiments/ENCSR176EXN/>

JUN ChIP-seq on human H1-hESC

GEO:GSM935614

<https://www.encodeproject.org/experiments/ENCSR000ECA/>

JUN ChIP-seq on human A549

GEO:GSE92221

<https://www.encodeproject.org/experiments/ENCSR996DUT/>

JUN ChIP-seq on human K562

GEO:GSM1003609

<https://www.encodeproject.org/experiments/ENCSR000EFS/>

H3K4me3 ChIP-seq on human NHLF

GEO:GSM733723

<https://www.encodeproject.org/experiments/ENCSR000AMW/>

H3K27ac ChIP-seq on human NHLF

GEO:GSM733646

<https://www.encodeproject.org/experiments/ENCSR000AMR/>

H3K9me3 ChIP-seq on human NHLF

GEO:GSM1003531

<https://www.encodeproject.org/experiments/ENCSR000ARQ/>

H3K27me3 ChIP-seq on human NHLF

GEO:GSM733764

<https://www.encodeproject.org/experiments/ENCSR000AMS/>

RNA-seq on IPF and normal lungs.

GSE52463

<https://www.ncbi.nlm.nih.gov/geo/query/acc.cgi?acc=GSE52463>

4. In the immunofluorescence presented in Figure 1H, the authors attempt to confirm the results obtained by mass cytometry. However, the authors did not present the results from co-immunostaining using antibodies specific for CD47 and PD-L1 and demonstrating the presence of these two proteins in 10% of lung fibroblasts. According to panel 1G they should detect around 10% of cells in which both proteins are present. In addition, the authors forgot to place the scale bars in panel 1H.

We appreciate this comment. We inserted the scale bars for all the IF images and quantified the immune stains (see **Fig.S1c** in the revised manuscript), which demonstrate comparable results to our CyTOF studies. Unfortunately, co-immune stains with antibodies for CD47 and PD-L1 are not possible, diagnostic antibodies are only available for the same species.

5. For the principal component analysis of lung fibroblasts after mass cytometry, the authors selected six markers: PD-L1, PD-L2, CD47, CALRETICULIN, FSP1, PODOPLANIN and PDGFRA. The author should explain in the text the criteria of selection of these markers, and whether other fibroblasts related markers were not included in this principal component analysis.

We thank the reviewer for this comment. Initially we decided to choose these markers because they appeared to highlight the profibrotic fibroblasts. However, we did repeat the PCA not only for the fibroblasts but also macrophages, now including all the markers. The repeated PCA separated IPF from normal lung as well.

6. The authors analyzed single cell suspensions of 14 representative lung samples (11 IPF samples and 3 control samples). The authors have to provide clinical data related to the samples used. For example: IPF stage, gender, age, smoking history, other pathologies, etc. If the IPF samples were from different disease stages, this might

explain the increase on the fibroblast percentage between 50 to 90% observed in the in figure 1C.

We thank the reviewer for this excellent suggestion and now include a table below as well as in supplemental materials specifying the pulmonary fibrosis stage and providing the requested additional clinical history. While all our pulmonary fibrosis patients were diagnosed on histology as end-stage pulmonary fibrosis with a severe diffusion defect (DLCO <30%) and clinically declining thus fulfilling criteria for transplant, there was variable disease involvement between 50-90% (by histology, CyTOF and CT). See supplemental table 1 of the manuscript.

We only included patients in our studies which had histologic or radiographic evidence of end-stage fibrosing interstitial lung disease (ILD): UIP (8), fibrotic NSIP (2), fibrotic chronic interstitial pneumonitis (2), DLCO all severe decreased DLCO between <20 - <30% of predicted ,FVC <80%, FVC 10% or greater decrement in FVC during 6-month follow-up, 6 minute walk pulse oximetry below 88% or 50m decline in over 6 months, ***associated pulmonary hypertensive (PAH) features on histopathology. # Our healthy control lung specimens were derived from lung lobectomy specimens for lung cancer, we only received histologic healthy appearing lung (lung specimen weights 150-200g, tumor diameters ranging 0.8-2cm, stage pT2pN0).

7. Is the observation from Figure 1I, claiming an increase of PD-L1 and CD47 expression in IPF when compared to control lungs, sustained by the expression profile in RNA-seq experiments published and used by the authors to confirm the ATAC-seq (see reference 35 and figure S3B)?

We thank the reviewer for this comment. The expression profile of PD-L1, CD47, IL6 and JUN from published RNA-seq data was listed below. The change was not always consisted with our data. But please noticed that this is bulk RNA-seq data which all the different cell lineages are included. Our finding is based on specific cell type

“fibroblasts”. And our ATAC-seq has been confirmed by our JUN ChIP-seq data and histone ChIP-seq data on line which we discussed in **Fig.4**.

gene	experiment	assay	tissue	pval	padj	log2fc	id
CD47	GSE52463	RNA-Seq	lung	3.41e-02	3.45e-01	-0.46	ENSG00000196776
JUN	GSE52463	RNA-Seq	lung	6.49e-02	4.76e-01	-0.73	ENSG00000177606
CD274	GSE52463	RNA-Seq	lung	1.72e-02	2.40e-01	-0.42	ENSG00000120217
IL6	GSE52463	RNA-Seq	lung	4.17e-02	3.88e-01	1.42	ENSG00000136244

As mentioned under point 3, the GEO accession number of the RNA-seq data has to be provided in the manuscript.

We apologize and now include the GEO accession number of the published RNA seq data here.

RNA-seq on IPF and normal lungs.

GSE52463

<https://www.ncbi.nlm.nih.gov/geo/query/acc.cgi?acc=GSE52463>

8. The authors conclude that suppressive immune cell types predominate in the IPF lung, but they might have preferential topology along the lung tissue. What would be the tissue infiltration rate from different pulmonary sections?

This is an excellent comment. We have analyzed lung tissues derived from different areas demonstrating variable involvement of pulmonary fibrosis by histomorphology; however to our big surprise samples from much less involved areas demonstrated about the same rate of immune infiltration with more involved areas from the same patient.

9. Several functional assays in primary fibroblasts (proliferation, migratory profile, Sircoll, Sirius Red, hydroxyproline, etc.) are required upon inhibition of CD47, PD-L1 and IL6.

We thank the reviewer for suggesting these experiments. We successfully demonstrated a significant reduction of fibroblasts and fibrosis after inhibition with CD47, PD-L1 and IL6 by Masson trichrome stains (which is not as sensitive but comparable to Sirius red and used for clinical assessment of lung fibrosis in Stanford) as well as immune staining for collagen to assess for extracellular matrix deposition; however since primary fibroblasts after treatment were quite rare we were not able to isolate sufficient numbers for extensive in vitro assays. However, we treated primary

fibroblast monocultures with a-CD47, a-PDL1 and a-IL-6 acutely in vitro and did not see any striking differences on proliferation and migration.

As suggested, we performed hydroxyproline assays in primary fibroblasts after treatment (Tx) with IL6, CD47 and HAC which demonstrated decreased hydroxyproline content compared to untreated (unTx).

10. The analysis demonstrating the statistical relevance of the data should be presented in the manuscript. P-values and the test used for the calculation should be presented in the figures and figure legends. In addition, the author might consider to summarize all these information related to all the data presented in the manuscript in a supplementary excel file.

We thank the reviewer for these comments. We now provide the statistical data organized by figures as a supplemental table 4.

Minor concerns:

11. Scale bars in all microscopy pictures should be presented.

We appreciate this comment, we now added scale bars for each panel instead of providing a single scale bar for all images with the same power.

12. Labelling of the figure panels is not uniform with respect of the lettering size. In addition, figure panels are too small to recognize the results

We thank the reviewer for this comment, and will provide uniform font size if a revised version of the manuscript will be accepted for publication.

13. References should be mentioned correctly. For example on page 3 “...While single cell deconvolution of the fibroblast heterogeneity was recently reported in a bleomycin initiated pulmonary fibrosis mouse model, no comprehensive single cell protein data focusing on fibroblast heterogeneity are yet available for human IPF (17).” The correct reference for this statement is the reference number 18.

We thank the reviewer #1 for catching this mistake and apologize for this oversight, we corrected the reference to 18.

14. On page 4 the authors wrote: “In addition to the increased abundance of fibroblasts, we performed a principal component analysis of the expression level of six markers (PD-L1, PD-L2, CD47, CALRETICULIN, FSP1, PODOPLANIN and PDGFRA) on fibroblasts and demonstrated that IPF lung fibroblasts from 13 IPF patients clustered together and were distinct from lung fibroblasts derived from normal healthy lungs (Figure 1D),...”. However, in the figure and in other part of the text is written that 11 IPF patients were analyzed.

Thanks for catching this oversight, the reviewer is correct we analyzed 11 pulmonary fibrosis patients and corrected the text on page 5.

15. The author should introduce abbreviations at the first place used, and implement them consequently along the whole manuscript.

Thanks for this comment to improve clarity and comprehension of our manuscript, we have now introduced all the abbreviations in supplementary table 5 and used consecutively.

16. The author should avoid repetition of the text thereby reducing redundancy.

We thank the reviewer for these stylistic suggestions, we carefully screened the text and eliminated redundancies in particular in the introduction and discussion part.

The data presented in the manuscript will be a major contribution for the field of IPF and of interest for scientists working on fibrosis in other organs. In addition, the manuscript has a strong translational potential suggesting therapeutic approaches against IPF. I strongly believe that after addressing all the concerns, the manuscript will improve significantly achieving the standards for publication at Nature Communications.

Reviewer #2

The manuscript submitted by Cui and colleagues entitled “Activation of JUN in fibroblasts promotes pro-fibrotic programme and dampens protective immunity” utilizes CyTOF to assess a panel of markers in immune cells and fibroblasts isolated from lung tissue of 11 humans with idiopathic pulmonary fibrosis (IPF) and 3 “healthy” controls. The authors report differences in the relative frequency of leukocytes and mesenchymal cells between the two groups and show that fibroblasts from the IPF subjects have increased expression of CD47, PDL1 and c-JUN. They also report increased expression of PD1 in macrophages and lymphocytes isolated from the IPF subjects. Using CRIPR-Cas9, the investigators delete JUN in IPF fibroblasts and show that this regulates CD47 enhancer activity and downstream expression of CD47. While the enhancer landscape of PDL1 was altered by JUN deletion in fibroblasts, downstream expression of PDL1 was not shown. Addition of IL-6 to fibroblast cultures increased CD47 expression. In an attempt to link PDL1, CD47 and IL-6 to fibrosis, the investigators blocked these pathways using antibodies, inhibitors and IL-6 knockout mice. Various combinations led to attenuated fibrosis. Lastly, the investigators employed a model in which human fibroblasts (IPF and normal) were transplanted into the kidney capsule of immunodeficient mice (Rag2ko mice lacking IL2R). Inhibition of PD1/PDL1 reduced fibroblast bioluminescence in the IPF group and attenuated the degree of fibrosis.

The experiments presented in the manuscript employ state-of-the-art technologies and leverage human tissue samples for mechanistic studies. Clearly, a huge amount of work

went into these studies. While the role of JUN as a driver of fibrosis is well recognized, the identification of JUN as a regulator of CD47 (and possibly PDL1 – although not fully shown) is novel. However there are a number of concerns with the manuscript as detailed below.

Major concerns.

1. The manuscript is difficult to follow and many key details are lacking (see below). It is difficult to follow many of the experiments and to know exactly how they were performed. Critical controls are lacking for some experiments. This leaves the reader with the impression that there was insufficient attention to detail.

We appreciate this comment; however, it is difficult to address this somewhat general statement without any specific critiques. Nevertheless, we critically reviewed and edited the entire manuscript and the figures and added as much detail as possible without losing conciseness. To improve the flow, we decompressed our figures now showing our data in 7 instead of previously 4 which now allows the reader to follow through more easily. In addition, we have completely revised our manuscript, deleted redundancies and added details to our method sections as well as detailed statistics which we show in supplemental table 4 organized by figures.

2. The manuscript has a number of pieces of data that aren't completely linked or explored. This leaves the manuscript somewhat scattered, and contributes to the overall difficulty with its interpretation. Moreover, many of the conclusions of the manuscript are overstated and not supported by sufficient experimental exploration. Just one example is CD47. The investigators provide solid evidence that fibroblasts from IPF patients have increased levels of CD47 and that JUN promotes CD47 expression. However, how CD47 contributes to fibrosis is not explored. For instance, would blockade of CD47 by itself dampen the fibrotic response? If so, how would this effect occur? Is expression of CD47 ligands (such as SIRP alpha) increased on other cells? One way to rescue the manuscript would be to trim it down and to tell a single concise story that is well supported by a full complement of experiments.

We thank the reviewer for these helpful comments and apologize for the lack of clarity in our previously submitted manuscript. We have now completely revised the manuscript addressing the reviewer comments and filling in the gaps. We find that CD47 blockade by itself dampens the fibrotic response. The mechanism of action is as following: CD47 highly expressed on fibroblasts interacts with its receptor SIRPa on macrophages thus preventing phagocytic removal and evading immune surveillance. We have analyzed SIRPa expression after fibrosis induction in mice with CyTOF and find that SIRPa expression is increased in CD45+ cells and in particular on macrophages. Blockage of CD47 with aCD47 antibody releases this protection and activates macrophages to remove these cells by phagocytosis.

3. The sources of tissue that comprise the IPF versus “healthy” samples are markedly different. This is likely to influence results.

We thank the reviewer for this important comment and agree with it, while we find excellent coverage for fibroblasts and immune cells, it is likely that we underestimated the content of epithelial and endothelial derived cells in our analysis; however, epithelial

and endothelial cells are not the main focus of our study, but fibroblasts and immune cells and we have excellent coverage for both cell types. We now include a table summarizing the clinical features of our patient derived specimens (supplementary table 1) which shows that the pulmonary fibrosis and control groups while from different sources are quite similar in composition. As a trained and practicing anatomical pathologist I am well aware of this not trivial problem of the proper “healthy” control lung tissue. We therefore tested different kinds of “healthy=normal lung samples to find the best “healthy” lung sample source. Since our IRB protocol does not allow for collection of healthy donor lung tissue during transplant (although the amount of lung tissue is rather small during that procedure), a simple source of “healthy lung tissue” was simply not available to us. We tried different sources including rapid autopsy lungs and ventilator lungs (with disappointing results) and came to the conclusion that VATS lobectomy lung specimens were quite similar in cell content to the areas of the pulmonary fibrosis lungs when both were sampled in the periphery, and we included only such specimens in our analysis.

a. The IPF tissue comes from explanted lungs whereas the “healthy” tissue comes from surgical resections (presumably this tissue is peri-tumoral). In the Methods section, the authors state that they tried to obtain post-mortem tissue but that viability was poor. While I empathize with the major challenges presented in obtaining “healthy” tissue, I am concerned that at least some of the results in Figs 1 and 2 may arise from this. (For instance, a marked absence in endothelial cells in “healthy” tissue and an absence of epithelial cells in IPF tissue).

The reviewer’s point is well taken. We already addressed part of this comment above. While it is likely that we underestimated epithelium and endothelium, there is no statistical significant difference between endothelial cells and epithelial cells between fibrotic and normal lung, both specimen types include comparable composition. We can explain the relative absence of endothelial and epithelial cells in both specimen types by peripheral sampling, which we also controlled by concurrent histopathologic sampling at time of tissue harvest.

b. Is it possible that some of the differences noted between the healthy and IPF tissues arises from the fact that the IPF tissue could contain larger airways and vessels whereas the “healthy” tissue will primarily contain alveolar regions. Fibroblasts, and other cell types may vary based on this fact alone.

We believe that we addressed this question already with our responses above.

c. The IPF versus “healthy” tissues should be more completely described in the methods. Of particular importance is knowing whether the “healthy” tissue samples were cryopreserved and whether the IPF samples were fresh (or also cryopreserved). The ages, sex, smoking status, etc should be given in a table (perhaps in the supplement).

We thank the reviewer for this insightful comment. All our samples (pulmonary fibrosis and normal) were harvested prospectively and processed immediately for CyTOF. We now provide a table in supplemental materials specifying all clinical information mentioned above (in the manuscript supplemental table 1).

d. BAL fluid from IPF and healthy controls are used. How were the BAL performed? Was the same volume instilled between groups and were returns comparable? Was the same technique used in both cases? Was the BAL from IPF subjects performed on post-mortem lung?

The BAL has been harvested from lungs of pulmonary fibrosis patients immediately (within 5 minutes) after explant, 5 mL were injected into the peripheral airspaces and at least 2 mL harvested for all specimens, which was subsequently snap frozen in liquid N₂. All specimens were surgical specimens and no post-mortem specimens were included.

4. The manuscript title suggests that JUN expression in fibroblasts dampens protective immunity. This is not shown experimentally.

We apologize for this overstatement and changed the title to modulate.

5. The ATAC seq studies suggest that PDL1 is downstream of JUN. This is not confirmed experimentally (as was nicely done with CD47). Along similar lines, the investigators suggest that IL-6 may enhance JUN expression / activity and that this will increase PDL1. Experimental evidence is lacking.

In this study, we found that the IL6 expression was upregulated after JUN induction in mice. We also quantified cytokines and chemokines contained in the supernatants of primary bone marrow, fibroblasts and monocyte cultures of JUN induced mice which showed increased IL6 in the culture supernatants suggesting that IL6 is secreted by BM and fibroblasts (**Fig.5**).

ATAC seq also suggested that *IL6* is downstream of *JUN*, and we detected increased IL-6 protein expression by flow with JUN overexpression (JUN-OE) which decreased with JUN deletion (JUN-KO).

Quite similar to *IL6*, the ATAC seq results had suggested that JUN could regulated PDL1. We did perform flow analysis for PDL1 protein expression based on the reviewer's comments which confirmed that the expression levels of *PDL1* increased after JUN overexpression and decreased after deletion.

6. Tissue immunohistochemistry results are not convincing. First, negative controls are not shown and it is possible that at least some of the signal arises from autofluorescence. Second, stains to show tissue architecture are lacking. H&E stains from serial sections (or at a minimum DIC images) would help orient the reader. Third, it is unclear whether the sections from the IPF subjects were taken at random or from areas with the greatest fibrosis. Since sections from 11 subjects were analyzed, some effort at quantification of the results should be performed.

Thanks for the comments. We do have performed the quantification at FigS1c and S2e. We thank the reviewer for this excellent suggestion and now have included H&E stains from serial sections of pulmonary fibrosis lung shown in Fig. S1 and Fig. S6 to help orient the reader. As mentioned by the reviewer there is a lot of autofluorescence in particular in the lung and we applied autofluorescence quenching software as specified

in Desai et al. *Elife* 2018⁷. We also include negative controls such as fibrotic and control lung sections stained with secondary antibody and or isotype control in every batch experiment, as well as fibroblasts and macrophages negative for CD47, PDL1 and PD1 respectively.

Additional comments.

The tissues from the IPF subjects appear to have a paucity of alveolar macrophages (AM) whereas the healthy control tissues have few, if any, interstitial macrophages (IM). Accordingly, the studies that compare macrophages from the IPF versus control may be influenced by markedly different ratios of the IM and AMs. An advantage to CyTOF is that the different cell types can be selected and compared head-to-head.

We thank the reviewer's excellent suggestions and agree that the advantage of CyTOF is to enable head-to-head comparisons within cell subsets. In response, we manually gated out AM and IM following the gating strategy of the previous report and performed tSNE analysis on AM and IM separately. Indeed, the immunophenotypes of AM and IM in the fibrotic tissues are also clearly different from those in the normal lungs. Accordingly, we have included all these results into main text and Fig S2.

The term “blood cells” is used throughout the manuscript and is quite confusing. The term “leukocyte is preferable”. Many of the cells that are isolated are likely to be outside of the circulation.

We thank the reviewer for this comment and now use leukocytes throughout the manuscript.

Gating strategies for the different cell types should be shown in the CyTOF experiments. This is especially true for macrophages and for fibroblasts. In terms of the latter, the investigators define fibroblasts as CD45- CD31- CK7- cells. It should be noted that this scheme would include other mesenchymal cells such as smooth muscle cells.

Thanks for the reviewer's comments. Yes, we define fibroblasts as CD45-CD31-CK7- cells since there is no universal fibroblast specific markers⁹. We have tried to use collagen1, collagen6, FAP, Vimentin, PDGFRa, PDGFRb, Desmin, FSP1, SMA and Sca1 to as the positive gate strategy to define fibroblast. However, the level of heterogeneity with these so-called fibroblast markers in CD31- CD45- CK7- population

could potentially bias our analysis if one single marker is selected to define fibroblasts. Furthermore, the mixed phenotype of certain fibroblasts, eg myofibroblasts, would also limit the extent of our analysis if other mesenchymal markers are included for negative gating. Hence, we decided to take the approach to define fibroblasts by only excluding leukocytes, epithelial cells, and endothelial cells.

The text confuses total cell numbers with the frequency of cells in several locations. I do not see that total cell numbers were ever evaluated for the tissue. An example of this occurs in the first paragraph of the Results section in which the authors state “we detected a 5-fold increase in fibroblasts...” It would be better to say that the frequency of fibroblasts was 5-fold higher in IPF tissues. Later in the same paragraph the text states “... fibroblasts in IPF lungs are not only increased in numbers...” While there are likely to be more fibroblasts in the IPF tissues, the data do not show this. They only show an increased percentage.

We thank the reviewer for this comment. To improve clarity, we changed the text to replace “we detected a 5-fold increase in fibroblasts...” with “the frequency of fibroblasts was 5-fold higher in fibrotic lung tissues”. We also corrected ourselves, instead of “fibroblasts in fibrotic lungs are not only increased in numbers” we state that fibroblasts are relative increased in their %.

Figure 1G. Presumably these are flow cytometry data (although not explicitly stated). It would be informative to show gating strategies used to arrive at the putative fibroblasts in the Supplement. In this regard, a panel with live-dead discrimination should be shown as well as FMO controls to ensure that changes in autofluorescence are not driving the putative differences.

We apologize for the confusion, these are CyTOF data, and were gated on lin-CD45-Epcam-CD31- and live cells. We now include a gating strategy as outlined above in supplemental figure 7. Due to the low background feature of CyTOF, FMO controls are

not applicable to CyTOF. Instead, the threshold for gating is decided by the background staining in the negative populations as we shown in the supplemental figure 7.

Figure 1J shows increased PDL1 protein in BAL from IPF lungs. What is the point of showing this? Is it assumed that PDL1 from the airspaces can gain access to the interstitium and have an effect on the fibroblasts?

We found it interesting that increased PDL1 protein is increased in BAL from fibrotic lungs could be an indirect evidence of PDL1 expression in the fibrotic lung. The mechanism and function of soluble PDL1 is beyond the scope of this study. To not distract the readers, we decided to leave it out from the manuscript.

Figure 2. The legend does not completely match the figure. This makes the figure very hard to interpret. In Figure 2A nearly 40% of cells are “other.” What are these cells? Can their identity be determined? If not, this brings the data from the rest of Figure 2A into question. Figure 2B doesn’t add much to the manuscript. Figure 2C and 2D are confusing. It is not clear what populations I and II are, nor is it clear in Figure 2D whether cells are from healthy lungs, IPF lungs, or both.

We apologize for this oversight: we now provide cell identity, these events are comprised of neutrophils, eosinophils and plasma cells.

Figure 3. It is curious that CD47 does not show up in Fig 3A. How do the authors explain this? Figs 3D and E demonstrate that Jun is upstream of CD47 in cultured fibroblasts. Within these experiments doxycycline is used to induce Jun expression. This effect of doxy is not mentioned anywhere in the text, and unless the reader is aware of the group’s previous work, these experiments are impossible to interpret. The text should be expanded to explain this.

We are grateful to the reviewer for catching this oversight, now we include the information for the CD47 promoter. We included in the method section that our JUN

overexpression is tet-on system, doxycycline is applied to turn on JUN overexpression. We tested for off target effects by doxycycline with Rosa rtta mice, which we treated with Dox and did not see any IL6 induction.

CD47 promoter coordinates											
chr	chrStart	chrEnd	promoter								
chr3	107807935	107811935	CD47								
Promoter length											
TSS											
Peaks overlapping CD47 promoter								IPF1KO1/IPF1wt1	IPF1KO2/IPF1wt2	Mean	
peaks ID	chr	chrStart	chrEnd	IPF1KO1	IPF1KO2	IPF1wt1	IPF1wt2	LogFC	LogFC	LogFC	p-value
CD47_1	chr3	107807543	107808043	12	6	32	13	0.375	0.461538462	0.41826923	0.0055
CD47_2	chr3	107809676	107810176	98	44	228	177	0.429824561	0.248587571	0.33920607	0.0183

Figure S4 shows IL-6 expression in a number of different cell types in culture. In these experiments c-JUN expression was reportedly induced with doxycycline. Is it possible that the increases in IL-6 are a result of off-target effects? It would be important to show that JUN was induced, and that inhibition of c-JUN (perhaps via an inhibitor) blocks IL-6.

To address the reviewer's comment, we now show that JUN was induced (**Fig. 4c, d**) and that inhibition of *JUN* blocks IL6 (**Fig. 5d**). We have carefully examined off target effects which we refer to in the comment below.

We performed similar experiments with a JUN inhibitor (T5224) and saw similar effects than with JUN deletion; JUN and IL6 decreased after inhibitor treatment.

We also tested for off target effects by doxycycline in Rosa rtta mice, which we treated with Dox and did not see any IL6 induction. In addition, we have published a manuscript Moore, Wernig, Longaker et al, 2018 in which we carefully investigated off target effects

of doxycycline and found mild anti-inflammatory effects as anticipated by an antibiotic drug, but no effects on IL6.¹⁰

And previous paper have characterized most elements of the IL-6 promoter that lie within the 300 bp proximal to the start site (+1) of transcription where multiple AP-1–binding sites were found.⁴

Kidney capsule studies with transplanted fibroblasts are intriguing, however more intense analysis would be reassuring. First, it appears that luminescence from the HAC-control fibroblasts is low at baseline. This is confusing, and suggests that there is a problem – perhaps low numbers of healthy fibroblasts were transplanted?

We thank the reviewer for this comment and agree, that transplanted fibroblasts are intriguing, however these experiments are quite challenging due to limited cell numbers etc. In addition, in our experience and also reported by others we find that primary fibroblasts lose their profibrotic nature in culture beyond 24 hours limiting their functional analysis.¹¹

To answer the reviewers question, identical viable cell numbers have been transplanted for both groups (normal and pulmonary fibrosis). We agree with the reviewer that the HAC-control fibroblasts at baseline are low initially, however we like to let the reviewer know that at this very early timepoint after virus infection protein expression can be quite variable (less than 16 hours) and that the expression of bioluminescence proteins becomes more stable; indeed we see bioluminescence for luciferase GFP is equal on Day 7 pretreatment for both groups.

Second, it is implied that fibroblasts from the HAC group are being removed by macrophages. Stains to show macrophages in the tissue would be helpful to support this – particularly if they show engulfment of fibroblasts.

We completely agree with the reviewer, since the fibroblasts reside in the kidney capsule of NSG mice which only have macrophages but lack B/T/NK cell, we imply that the fibroblasts under this circumstance are removed by macrophages, and this has been also functionally proven by Gordon et al. Nature 2017¹². While we attempted stains for macrophages, all the fibroblasts were already removed at this time point due to the highly effective HAC treatment.

However, we now have new additional data which demonstrate macrophage (labeled with *) engulfment with HAC treatment (see below).

Finally, it would be instructive to know if the IPF fibroblasts are dying. This could be achieved by staining the tissues.

Thanks for the comment. While we did not directly assess cell viability, we stained for Edu before transplant and there was no difference between the groups. While the fibroblasts appear viable on the untreated kidney transplants from the morphology by trichrome stain, we did not assess cell viability directly. In addition, we analyzed all other organs of these mice and did not detect GFP labeled human fibroblasts anywhere else, excluding the possibility of them migrating out.

Reviewer #4 (Remarks to the Author):

The manuscript from Cui et al. clearly describes a mechanism for the evolution of a fibrotic program in the lung. Experiments try to logically dissect the mechanism and overall gave strong evidence that JUN activation is key for the pro-fibrotic program. The use of data generated in humans and the final demonstration of the mechanism in a mouse model strongly support the hypothesis.

Could the authors make available the CyTOF FCS files?

Yes, it's attached in the supplement materials.

In Figure 2 A and B the title "Blood Cells" seems to indicate that the cells are derived from blood whereas these are cell collected from the lungs

We apologize for the confusing terminology, we changed to leukocytes throughout the manuscript.

Figure 2C show a tSNE analysis whereas in the text (page 6, line7) and figure legend is referred to as PCA. The legend of Figure 2F state that the Figure shows IDO expression whereas it shows the PCA. The Authors need to revise the entire legend and the references in the text.

We thank the reviewer for picking up this oversight. We have now corrected this issue and revised the legend and the references in the text.

The separation of single cells in the tSNE plots in Figure 2D is really poor. Will be possible to run the algorithm with more iterations or change the perplexity? I have the same observation for some of the other tSNE plots throughout the manuscript

Thanks for the comment. The new tSNE plots have been put in the **Fig. 2** and **Supplementary Fig. S2**.

It will be interesting to show in Supplementary Figure 2 the expression of CD47

Thanks for the comment. The expression of CD47 has been put in the **Supplementary Fig. S2**.

Could the Authors add a reference for the involvement of PDGF-BB and CCL5 in connective tissue remodeling (page 8)?

We included the above mentioned references:

PDGF-BB induces PRMT1 expression through ERK1/2 dependent STAT1 activation and regulates remodeling in primary human lung fibroblasts.

PMID: 26795953

Combination of roflumilast with a beta-2 adrenergic receptor agonist inhibits proinflammatory and profibrotic mediator release from human lung fibroblasts.

PMID:22452977

Why the Authors finally use the Bleomycin mouse model instead of the JUN-inducible mouse model for the checkpoint blockade experiments?

When we first published our JUN inducible fibrosis model we did receive lots of questions since people were not familiar with our model but they were with the bleomycin model. We subsequently discovered that with bleomycin induced fibrosis JUN also gets induced and activated. We used the bleomycin model out of poor practicability to be able to compare the same genetic background to the IL6KO mice since the bleomycin treated mice are C57/BL6 and similar to the IL6KO mice, but our JUN inducible mice are not fully backcrossed.

Could the author build a Supplementary table summarizing the number of living cells found in each sample and the number of cells used to run each algorithm?

Yes, the living cell numbers and cell number for tSNE has been summarized in supplemental table 3.

Supplementary Table 4: please indicate the fluorochromes for each antibody

The fluorochromes are added in the supplementary table 2, also see below.

Antigen	Vendor	Clone	Staining Concentration	fluorochromes
Collagen1	Abcam	ab34710	1:100	Alexa Fluor 594
Actin, Muscle (SMA)	Smooth marque	cell 1A4	1:200	Alexa Fluor 488
CD47	Abcam	B6H12.2	1:50	Alexa Fluor 488
PDL1	CST	E1L3N	1:100	Alexa Fluor 594
	Cell			
PD1	marque	NAT105	1:100	Alexa Fluor 594
PD1	Abcam	EPR48772	1:100	Alexa Fluor 488
CD68	Dako	KP1	1:200	Alexa Fluor 488
CD3	Abcam	ab5690	1:100	Alexa Fluor 594
PDL1	Abcam	MIH6	1:100	Alexa Fluor 488
FSP1	EMD	07-2274	1:200	Alexa Fluor 594
GFP	Abcam	ab13970	1:100	Alexa Fluor 488

In the Methods section, MIBI technology is not described

We apologize for this oversight and now include MIBI in the methods section and here.

Methods: For MIBI, slides were postfixated for 5 min (PBS, 2% glutaraldehyde), rinsed in dH₂O and stained with Hematoxylin for 10 s, at the end, the sections were dehydrated via graded ethanol series, air dried using a vacuum desiccator for at least 24 h before imaging. MIBI imaging are performed by NanoSIMS 50L spectroscopy (Cameca,

France) at Stanford Nano Shared Facilities (SNSF) and analyzed by using Image with Plugin OpenMIMS (NRIMS, <http://www.nrim.s.hms.harvard.edu>).

Reference:

- 1 Vierbuchen, T. et al. AP-1 Transcription Factors and the BAF Complex Mediate Signal-Dependent Enhancer Selection. *Mol Cell* 68, 1067-1082 e1012, doi:10.1016/j.molcel.2017.11.026 (2017).
- 2 Madrigal, P. & Alasoo, K. AP-1 Takes Centre Stage in Enhancer Chromatin Dynamics. *Trends Cell Biol* 28, 509-511, doi:10.1016/j.tcb.2018.04.009 (2018).
- 3 Coelho, M. A. et al. Oncogenic RAS Signaling Promotes Tumor Immuno-resistance by Stabilizing PD-L1 mRNA. *Immunity* 47, 1083-1099 e1086, doi:10.1016/j.immuni.2017.11.016 (2017).
- 4 Sitaraman, S. V. et al. Neutrophil-epithelial crosstalk at the intestinal luminal surface mediated by reciprocal secretion of adenosine and IL-6. *J Clin Invest* 107, 861-869, doi:10.1172/JCI11783 (2001).
- 5 Ridker, P. M., Rifai, N., Stampfer, M. J. & Hennekens, C. H. Plasma concentration of interleukin-6 and the risk of future myocardial infarction among apparently healthy men. *Circulation* 101, 1767-1772, doi:10.1161/01.cir.101.15.1767 (2000).
- 6 Gustems, M. et al. c-Jun/c-Fos heterodimers regulate cellular genes via a newly identified class of methylated DNA sequence motifs. *Nucleic Acids Res* 42, 3059-3072, doi:10.1093/nar/gkt1323 (2014).
- 7 Nagendran, M., Riordan, D. P., Harbury, P. B. & Desai, T. J. Automated cell-type classification in intact tissues by single-cell molecular profiling. *Elife* 7, doi:10.7554/eLife.30510 (2018).
- 8 Bharat, A. et al. Flow Cytometry Reveals Similarities Between Lung Macrophages in Humans and Mice. *Am J Respir Cell Mol Biol* 54, 147-149, doi:10.1165/rcmb.2015-0147LE (2016).
- 9 Martinez, F. J. et al. Idiopathic pulmonary fibrosis. *Nat Rev Dis Primers* 3, 17074, doi:10.1038/nrdp.2017.74 (2017).

- 10 Moore, A. L. et al. Doxycycline Reduces Scar Thickness and Improves Collagen Architecture. *Ann Surg*, doi:10.1097/SLA.0000000000003172 (2018).
- 11 Rinkevich, Y. et al. Skin fibrosis. Identification and isolation of a dermal lineage with intrinsic fibrogenic potential. *Science* 348, aaa2151, doi:10.1126/science.aaa2151 (2015).
- 12 Gordon, S. R. et al. PD-1 expression by tumour-associated macrophages inhibits phagocytosis and tumour immunity. *Nature* 545, 495-499, doi:10.1038/nature22396 (2017).

Reviewers' comments:

Reviewer #1 (Remarks to the Author):

Peer review of the manuscript with the number NCOMMS-19-12042A at Nature Communications by Cui L, Chen SY et al and with the title: "Activation of JUN in fibroblasts promotes pro-fibrotic programme and modulate protective immunity"

The authors addressed the majority of the major concerns raised by the reviewers. For those concerns that were not experimentally addressed, the authors provided justification in terms of limited access to material, such as primary cells or antibodies, and in some cases they provided other options. Only for one of the concerns raised by me (Reviewer #1), the authors claimed that it was out of the scope of the study, which is a subjective statement, since I still consider that this experiment would have further increased the significance of their work. However, the authors have enriched their conclusions on chromatin accessibility obtained by ATAC-seq by repeating this approach also in wild-type lung fibroblasts with or without JUN gain-of-function, obtaining consistent results compared to the JUN loss-of-function in fibrotic fibroblasts and allowing a genome-wide analysis of the regulation mediated by JUN. One of the most important improvements to the manuscript was the use of a combinatorial analysis of ATAC and JUN ChIP-seq to evaluate the JUN binding to genomic regions of interest in primary fibroblasts and its correlation with the accessible chromatin state. Despite the limited material obtained from primary cells, the authors added new data present in several main and supplementary figures including ChIP-seq data using the JUN antibody, as well as histone marks associated to open (H3K4me3 and H3K27ac) or closed (H3K9me3, H3K27me3) chromatin by ChIP-seq data retrieved from the GEO database, which is indeed supporting their conclusions and gives more relevance to their findings. The new analysis was performed not only through JUN loss-of-function in fibrotic lung fibroblasts but also by consistent gain-of-function conditions in normal lung fibroblasts, thereby demonstrating the direct effect of JUN in its own promoter accessibility, as well as in other fibrotic genes. Thus, in the revised version of the manuscript the authors can certainly conclude by ATAC-seq and JUN ChIP-seq combinatorial analysis on normal primary fibroblasts that JUN binds close to IL6 promoter regions and increases its promoter accessibility when JUN is overexpressed, which explains the observed increase on IL6 expression by qRT-PCR and IL6 protein levels by WB. Remarkably, the analysis performed during the manuscript revision potentiate the role of JUN as a key factor with chromatin remodeling effects, since its occupancy was observed at published super-enhancer regions of the target genes CD47 and PD-L1.

The revised version of the manuscript confirms my previous opinion about the novelty of the work and the impact that this work will have to the IPF field. Conceptually, the characterization of the heterogeneous population of cells that is relevant for the pathophysiology of human idiopathic pulmonary fibrosis is novel and certainly one of the strengths of the manuscript. While single cell

deconvolution of the fibroblast heterogeneity was recently reported in a bleomycin mouse model for pulmonary fibrosis, no comprehensive single cell data are yet available for human IPF. The data presented by the authors will be a major contribution to the IPF field and also of interest for scientists working on fibrosis of other organs. The concept of JUN being a master regulator of lung fibrosis in IPF is not new. The authors demonstrated in one of their previous publications that activation of JUN in a transgenic mouse model is sufficient to induce lung fibrosis. However, the proposed mechanism of JUN mediating changes in chromatin structure that regulates the expression of pro-fibrotic genes and immune checkpoint pathways genes is in turn novel. Other strength of the manuscript is that the authors used elaborated and of state-of-the-art methods for obtaining the data supporting their claims. The clinical relevance of the manuscript is reflected in the data that support a therapeutic approach against IPF based on a triple combined PD-L1, CD47 and IL6 blockade.

Summarizing, I recognize the excellent work performed by the authors addressing most of the concerns of the different reviewers, thereby improving significantly not only the structure of the manuscript, but also obtaining additional data that strengthen their conclusions. I recommend the publication of the revised manuscript after addressing the following minor concerns:

Minor concerns:

1. The authors has to double-check grammar and redaction mistakes. For example

1.1 in the title the word “modulate” should be substituted by “modulates”

1.2 in line 76, the word “identify” has to be changed to “identity”

1.3 in line 327, the authors typed two times the same words “This outcome is confirmed this outcome by loss...”

I recommend a revision of the manuscript by a native English speaker, which should detect these and other details.

2. Please double check the scale bars in the microscopy pictures. For example, in the Figures 2g and 3g the scale bars should represent 100um. However the extreme difference in the size of the cells presented in the pictures suggests that the scale bars in both Figures represent different sizes.

Reviewer #2 (Remarks to the Author):

The manuscript by Cui et al is significantly improved. The investigators have added a number of additional experiments that help round out the manuscript. Additionally, they have clarified some critical points and improved presentation of the figures. However, some concerns linger. None of these specifically require additional experiments.

Major

There are multiple grammatical errors throughout the text. Some sections remain difficult to understand. In addition, some results are overstated or not interpreted correctly. A careful re-read of the paper by all of the authors is essential. I have highlighted a few of the errors in the Minor Comments below.

Figure 1H is somewhat confusing. I believe the goal is to show that CD47 co-localizes with fibroblasts. However, since collagen is used as a co-stain, this simply shows the existence of CD47 positive cells in areas where fibrosis is present. Co-localization with fibroblasts is not achieved. In addition, the CD47 staining is not convincing. It is very difficult to see in the images provided.

Figure 2G is unchanged from the original version of the manuscript and remains problematic for two reasons. First, the authors claim that this set of images shows that PD1 is increased on macrophages in fibrotic lungs. The figure doesn't really show this. It shows that there are more macrophages in the fibrotic lungs. The intensity of PD1 staining seems the same in fibrotic vs healthy lungs. Second, all macrophages in the images appear to be PD1+. There are none that are PD1 negative. This is at odds with Figure S2f which shows that only 10% of macrophages in the fibrotic lungs are PD1+ and that less than 1% are positive in the healthy lungs.

In the initial review it was pointed out that none of the figures show total cell numbers (i.e. absolute numbers of cells). They all show percentages. Hence it is important to describe the data in terms of ratios or percentages of cells and not total numbers. While some instances have been corrected, others still exist. The authors are strongly encouraged to be precise in this point throughout the manuscript. An example of this lingering issue can be found in Lines 174-180. The authors state that absolute T cells are increased (presumably in fibrotic lungs, line 174) and that increased numbers of regulatory T cells (line 179) and exhausted T cells are (line 180) are present in fibrotic lungs. The data

show that the PERCENTAGES of these subsets are increased, not that total numbers are increased. Of note, the corresponding figure legend also falls victim to this pitfall (Lines 775, 776, 779) as do lines 294, 295

Figure 3D. The authors suggest that this shows “increased PD-1+ expression on T cells” Presumably this statement relates to fibrotic lungs, although not expressly stated. The problem is that the figure doesn’t show increased expression of PD-1 on T cells in fibrotic lungs. It shows that a greater percentage of the T cells in the fibrotic lungs express some PD-1. A second problem with the images shown is that some of the T cells in the healthy lung are intensely green (i.e. very high expression of PD1) – much more so than in the fibrotic lung. This contradicts the interpretation of results in the text.

Phagocytosis experiments are superficial and don’t add to the paper. The authors should consider removing them. First, the methods are not clear. The authors reference a paper in which they performed this assay, but complete details are not provided in that paper. Second, in the manuscript under review, details are also lacking. The concentrations and timing of the HAC protein and anti-CD47 Ab are not provided. It is important to note if the macrophages were pre-treated, or if the antibody was added to co-culture. Addition of the antibody to the targets would mask CD47 and increase Fc mediated uptake of the cells through opsonization. Any antibody that binds to fibroblasts would do this. Third, the flow cytometry technique that is used will not distinguish cell binding from true engulfment. Fourth, were the fibroblasts that were presumed to be engulfed apoptotic, necrotic or neither. Additional experiments would be needed to round out the phagocytosis experiments. Since they don’t add to the overall message of the paper, they should be eliminated.

Minor Comments

Line 45. The generic names for the drugs are appropriately provided. However, they should not be capitalized (since they are generic). The same is true for azathioprine later in the text.

Line 45. Pirfenidone is misspelled in the text (needs an “e” at the end.)

Line 45-46 suggests that pirfenidone targets growth factor receptors. This is not correct. To my knowledge the receptors are not targeted.

Lines 45-47 imply that nintedanib does not have “long-term disease modifying effects in pulmonary fibrosis.” However, just in the last week a new study came out in the New England Journal (N Engl J Med 2019; 381:1718-1727) that shows a benefit in lung function at one year. The manuscript would benefit from this update.

Line 110 indicates that CK7 identifies “broncho epithelial cells.” Doesn’t it also identify alveolar epithelial cells? If this is the case, then simply referring to the cells as epithelial cells should be fine.

Lines 124-5. Why are the words PODOPLANIN and CALRETICULIN in capitals? These molecules should be in lower case.

Line 140. Please revise to indicate that a third of the fibroblasts upregulates EITHER ONE of two immune checkpoint proteins: CD47 and PD-L1. Dual upregulation is only demonstrated in 9%.

Line 149-150 states “in fibrotic compared to normal lung controls, we observed no significant differences in T cell and macrophage numbers.” This is not what the data show. There is no difference in the percentage of the cells” Absolute number of cells are not measured here.

Line 153-154 The following sentence doesn’t make sense. Please revise. “However, recently, there are growing number of literatures have focus on the complexity of the lung myeloid, which also exists another subsets of dendritic cells, tissue monocytes and nonalveolar macrophages, called interstitial macrophages.”

Lines 275-277 read “In this paper, we have made the intriguing observations that two critical immune check point proteins CD47 and PD-L1 – are not only induced in two mouse models of lung fibrosis (JUN and bleomycin mediated), but also in lung fibroblasts of human pulmonary fibrosis” This is confusing. Are the authors referring to another paper that needs to be referenced or are they referring to the current work. If they are referring to the current work, the reader will be taken off guard since data have not yet been presented using bleomycin treated mice.

Line 286 refers to “bronchoepithelial cells”. EPCAM also identifies alveolar epithelial cells. Hence, it seems “epithelial cells” is the correct designation.

Line 291 and elsewhere. The phrase “PD-L1 ligand” is redundant. The “L” in PD-L1 stands for ligand.

Line 305 has been added and states that FSP-1 is highly specific for fibroblasts. This is incorrect. Multiple reports show that it is not specific (see PMID: 23997102, 21173249 as examples).

Figure 1 H has an arrow in the SMA panel for the normal lung. Why is this there?

Figure 7 Legend is poorly written and difficult to understand. The first sentence is over 60 words in length and should be broken into at least two sentences if not more.

Reviewer #4 (Remarks to the Author):

The Authors answered all my questions and from my side, I do not have further comments.

Reviewers' comments:

Reviewer #1 (Remarks to the Author):

Peer review of the manuscript with the number NCOMMS-19-12042A at Nature Communications by Cui L, Chen SY et al and with the title: "Activation of JUN in fibroblasts promotes pro-fibrotic programme and modulate protective immunity"

The authors addressed the majority of the major concerns raised by the reviewers. For those concerns that were not experimentally addressed, the authors provided justification in terms of limited access to material, such as primary cells or antibodies, and in some cases they provided other options. Only for one of the concerns raised by me (Reviewer #1), the authors claimed that it was out of the scope of the study, which is a subjective statement, since I still consider that this experiment would have further increased the significance of their work. However, the authors have enriched their conclusions on chromatin accessibility obtained by ATAC-seq by repeating this approach also in wild-type lung fibroblasts with or without JUN gain-of-function, obtaining consistent results compared to the JUN loss-of-function in fibrotic fibroblasts and allowing a genome-wide analysis of the regulation mediated by JUN. One of the most important improvements to the manuscript was the use of a combinatorial analysis of ATAC and JUN ChIP-seq to evaluate the JUN binding to genomic regions of interest in primary fibroblasts and its correlation with the accessible chromatin state. Despite the limited material obtained from primary cells, the authors added new data present in several main and supplementary figures including ChIP-seq data using the JUN antibody, as well as histone marks associated to open (H3K4me3 and H3K27ac) or closed (H3K9me3, H3K27me3) chromatin by ChIP-seq data retrieved from the GEO database, which is indeed supporting their conclusions and gives more relevance to their findings. The new analysis was performed not only through JUN loss-of-function in fibrotic lung fibroblasts but also by consistent gain-of-function conditions in normal lung fibroblasts, thereby demonstrating the direct effect of JUN in its own promoter accessibility, as well as in other fibrotic genes. Thus, in the revised version of the manuscript the authors can certainly conclude by ATAC-seq and JUN ChIP-seq combinatorial analysis on normal primary fibroblasts that JUN binds close to IL6 promoter regions and increases its promoter accessibility when JUN is overexpressed, which explains the observed increase on IL6 expression by qRT-PCR and IL6 protein levels by WB. Remarkably, the analysis performed during the manuscript revision potentiates the role of JUN as a key factor with chromatin remodeling effects, since its occupancy was observed at published super-enhancer regions of the target genes CD47 and PD-L1.

The revised version of the manuscript confirms my previous opinion about the novelty of the work and the impact that this work will have to the IPF field. Conceptually, the characterization of the heterogeneous population of cells that is relevant for the pathophysiology of human idiopathic pulmonary fibrosis is novel and certainly one of the strengths of the manuscript. While single cell deconvolution of the fibroblast heterogeneity was recently reported in a bleomycin mouse model for pulmonary fibrosis,

no comprehensive single cell data are yet available for human IPF. The data presented by the authors will be a major contribution to the IPF field and also of interest for scientists working on fibrosis of other organs. The concept of JUN being a master regulator of lung fibrosis in IPF is not new. The authors demonstrated in one of their previous publications that activation of JUN in a transgenic mouse model is sufficient to induce lung fibrosis. However, the proposed mechanism of JUN mediating changes in chromatin structure that regulates the expression of pro-fibrotic genes and immune checkpoint pathways genes is in turn novel. Other strength of the manuscript is that the authors used elaborated and of state-of-the-art methods for obtaining the data supporting their claims. The clinical relevance of the manuscript is reflected in the data that support a therapeutic approach against IPF based on a triple combined PD-L1, CD47 and IL6 blockade.

Summarizing, I recognize the excellent work performed by the authors addressing most of the concerns of the different reviewers, thereby improving significantly not only the structure of the manuscript, but also obtaining additional data that strengthen their conclusions. I recommend the publication of the revised manuscript after addressing the following minor concerns:

Minor concerns:

1. The authors has to double-check grammar and redaction mistakes. For example
 - 1.1 in the title the word “modulate” should be substituted by “modulates”
 - 1.2 in line 76, the word “identify” has to be changed to “identity”
 - 1.3 in line 327, the authors typed two times the same words “This outcome is confirmed this outcome by loss...”I recommend a revision of the manuscript by a native English speaker, which should detect these and other details.

We are grateful for reviewer 1 comments recognizing the impact of our study. We corrected the spelling errors outlined, in addition a native English speaker has revised grammar mistakes.

2. Please double check the scale bars in the microscopy pictures. For example, in the Figures 2g and 3g the scale bars should represent 100um. However the extreme difference in the size of the cells presented in the pictures suggests that the scale bars in both Figures represent different sizes.

We thank the reviewer for picking up on this error which we now corrected!

Figure 2

Figure 3

Figure 2. (g) Representative images of immune fluorescent stains highlighted increased PD-1 expression on macrophages from fibrotic lung tissues (Scale bars, 100 μ m).

Figure 3. (g) Representative images of immune fluorescent stains for PD-1 on T cells (CD3) highlighting increased percentages of PD-1+ T cells in fibrotic lung samples (Scale bars, 100 μ m).

Reviewer #2 (Remarks to the Author):

The manuscript by Cui et al is significantly improved. The investigators have added a number of additional experiments that help round out the manuscript. Additionally, they have clarified some critical points and improved presentation of the figures. However, some concerns linger. None of these specifically require additional experiments.

Major

There are multiple grammatical errors throughout the text. Some sections remain difficult to understand. In addition, some results are overstated or not interpreted correctly. A careful re-read of the paper by all of the authors is essential. I have highlighted a few of the errors in the Minor Comments below.

We appreciate this comment. A native speaker has now revised the manuscript and eliminated grammar errors, we also have carefully revised the manuscript and corrected the issues outlined by reviewer #2.

Figure 1H is somewhat confusing. I believe the goal is to show that CD47 co-localizes with fibroblasts. However, since collagen is used as a co-stain, this simply shows the existence of CD47 positive cells in areas where fibrosis is present. Co-localization with fibroblasts is not achieved. In addition, the CD47 staining is not convincing. It is very difficult to see in the images provided.

We thank the reviewer for this statement and completely agree, the extensive extracellular matrix present makes specific cellular collagen staining pretty difficult to discriminate. We include stains demonstrating double-labeling fibroblastic cells in this area with SMA/FSP1 and CD47 (low and high-power confocal images). In addition, we show MIBI images demonstrating co-localization of collagen with CD47 on fibroblasts without labeling background matrix.

Figure 2G is unchanged from the original version of the manuscript and remains problematic for two reasons. First, the authors claim that this set of images shows that PD1 is increased on macrophages in fibrotic lungs. The figure doesn't really show this. It shows that there are more macrophages in the fibrotic lungs. The intensity of PD1 staining seems the same in fibrotic vs healthy lungs. Second, all macrophages in the images appear to be PD1+. There are none that are PD1 negative. This is at odds with Figure S2f which shows that only 10% of macrophages in the fibrotic lungs are PD1+ and that less than 1% are positive in the healthy lungs.

The reviewer's comment is well taken. While overall PD1 expression is only present in a small subset of macrophages, we find that they are clustering in

focal patches as displayed. However, in general there are many fewer macrophages expressing PD1+ therefore the reviewer is correct that the original image suggests a much higher co-expression of PD-1 in macrophages. We revisited our IPF samples and took more representative images, which we now include (please also see below) reflecting the full spectrum of PD-1 expression in macrophages in end-stage fibrosing interstitial lung disease.

Figure 2

In the initial review it was pointed out that none of the figures show total cell numbers (i.e. absolute numbers of cells). They all show percentages. Hence it is important to describe the data in terms of ratios or percentages of cells and not total numbers. While some instances have been corrected, others still exist. The authors are strongly encouraged to be precise in this point throughout the manuscript. An example of this lingering issue can be found in Lines 174-180. The authors state that absolute T cells are increased (presumably in fibrotic lungs, line 174) and that increased numbers of regulatory T cells (line 179) and exhausted T cells (line 180) are present in fibrotic lungs. The data show that the PERCENTAGES of these subsets are increased, not that total numbers are increased. Of note, the corresponding figure legend also falls victim to this pitfall (Lines 775, 776, 779) as do lines 294, 295

We thank the reviewer again for picking up on this imprecision which we corrected for lines 174-180. Increased percentages of T cells including increased percentages of regulatory and exhausted T cells are present in the fibrotic lungs.

Figure 3D. The authors suggest that this shows “increased PD-1+ expression on T cells” Presumably this statement relates to fibrotic lungs, although not expressly stated. The problem is that the figure doesn’t show increased expression of PD-1 on T cells in fibrotic lungs. It shows that a greater percentage of the T cells in the fibrotic lungs express some PD-1. A second problem with the images shown is that some of the T cells in the healthy lung are intensely green (i.e. very high expression of PD1) – much more so than in the fibrotic lung. This contradicts the interpretation of results in the text.

Figure 3D. We thank the reviewer and agree. We changed the wording to: It shows that a greater percentage of the T cells in the fibrotic lungs express PD-1. PD-1

expression CD8 T cells present in normal lung is very rare as shown in Fig.2f; however we do notice PD-1 expression in a small subset of CD4 T cells present in normal lung as indicated in Fig.S3d. We completely agree with the reviewer on the 2nd aspect, and now show more representative images of stains with similar exposure time demonstrating equal bright PD1 expression in healthy lungs.

Figure 3

Phagocytosis experiments are superficial and don't add to the paper. The authors should consider removing them. First, the methods are not clear. The authors reference a paper in which they performed this assay, but complete details are not provided in that paper. Second, in the manuscript under review, details are also lacking. The concentrations and timing of the HAC protein and anti-CD47 Ab are not provided. It is important to note if the macrophages were pre-treated, or if the antibody was added to co-culture. Addition of the antibody to the targets would mask CD47 and increase Fc mediated uptake of the cells through opsonization. Any antibody that binds to fibroblasts would do this. Third, the flow cytometry technique that is used will not distinguish cell binding from true engulfment. Fourth, were the fibroblasts that were presumed to be engulfed apoptotic, necrotic or neither. Additional experiments would be needed to round out the phagocytosis experiments. Since they don't add to the overall message of the paper, they should be eliminated.

GW: Thanks for pointing this out. We completely agree with the reviewer in all aspects and removed these data.

Minor Comments

Line 45. The generic names for the drugs are appropriately provided. However, they should not be capitalized (since they are generic). The same is true for azathioprine later in the text.

Thanks for pointing this out, we corrected this here and also for azathioprine.

Line 45. Pirfenidone is misspelled in the text (needs an "e" at the end.)

Thanks, we corrected to pirfenidone.

Line 45-46 suggests that pirfenidone targets growth factor receptors. This is not correct. To my knowledge the receptors are not targeted.

We thank the reviewer for picking up on this. Nintedanib targets the receptor tyrosine kinases VEGFR, FGFR and PDGFR. Pirfenidone is a small molecule inhibiting the TGFB pathway.

Lines 45-47 imply that nintedanib does not have “long-term disease modifying effects in pulmonary fibrosis.” However, just in the last week a new study came out in the New England Journal (N Engl J Med 2019; 381:1718-1727) that shows a benefit in lung function at one year. The manuscript would benefit from this update.

Line 45-47: We appreciate this comment by the reviewer. We updated the reference. Also, we intended to say that none of the current treatments are curative, and corrected this in the introduction.

Line 110 indicates that CK7 identifies “broncho epithelial cells.” Doesn’t it also identify alveolar epithelial cells? If this is the case, then simply referring to the cells as epithelial cells should be fine.

Line 110: Thanks, we deleted broncho epithelial corrected to epithelial cells.

Lines 124-5. Why are the words PODOPLANIN and CALRETICULIN in capitals? These molecules should be in lower case.

Lines 124-5. Thanks for pointing out this, we changed them.

Line 140. Please revise to indicate that a third of the fibroblasts upregulates EITHER ONE of two immune checkpoint proteins: CD47 and PD-L1. Dual upregulation is only demonstrated in 9%.

Thanks for pointing this out, we revised line 140: "A third of the fibroblasts upregulate either one of two immune checkpoint proteins, CD47 and PD-L1, and dual upregulation is demonstrated in ~9%."

Line 149-150 states “in fibrotic compared to normal lung controls, we observed no significant differences in T cell and macrophage numbers.” This is not what the data show. There is no difference in the percentage of the cells” Absolute number of cells are not measured here.

We thank the reviewer for this comment and corrected our statement to “in fibrotic lungs compared to normal control lungs, we observed no significant difference in the percentages of T cells and macrophages.”

Line 153-154 The following sentence doesn’t make sense. Please revise. “However, recently, there are growing number of literatures have focus on the complexity of the lung myeloid, which also exists another subsets of dendritic cells, tissue monocytes and nonalveolar macrophages, called interstitial macrophages.”

We thank the reviewer for picking up on this, we corrected as following: “A number of recent reports have focused on the complexity of the myeloid cells present in the lung and provide evidence that interstitial macrophages are a heterogenous population comprised dendritic cells, tissue monocytes and nonalveolar macrophages”.

Lines 275-277 read “In this paper, we have made the intriguing observations that two critical immune check point proteins CD47 and PD-L1 – are not only induced in two mouse models of lung fibrosis (JUN and bleomycin mediated), but also in lung fibroblasts of human pulmonary fibrosis” This is confusing. Are the authors referring to another paper that needs to be referenced or are they referring to the current work. If they are referring to the current work, the reader will be taken off guard since data have not yet been presented using bleomycin treated mice.

Lines 275-277: We apologize for the confusion and corrected as following: " In this paper, we have made the intriguing observations that two critical immune check point proteins - CD47 and PD-L1 – are not only induced in a mouse models of lung fibrosis, but also in lung fibroblasts of human pulmonary fibrosis”.

Line 286 refers to “bronchoepithelial cells”. EPCAM also identifies alveolar epithelial cells. Hence, it seems “epithelial cells” is the correct designation.

Line 286: We thank the reviewer for this comment, we provide now the correct designation to "epithelial cells"

Line 291 and elsewhere. The phrase “PD-L1 ligand” is redundant. The “L” in PD-L1 stands for ligand.

Thanks for pointing out the redundancy, we eliminated ligand throughout the manuscript.

Line 305 has been added and states that FSP-1 is highly specific for fibroblasts. This is incorrect. Multiple reports show that it is not specific (see PMID: 23997102, 21173249 as examples).

Thanks for pointing this out, the reviewer is absolutely correct, FSP-1 is not specific to fibroblasts, but appears to be a good marker for pathologic lung fibroblasts in lung fibrosis in the absence of any better and specific fibroblast marker, please find reference included here and the manuscript.

Lawson, W. E. et al. Characterization of fibroblast-specific protein 1 in pulmonary fibrosis. American journal of respiratory and critical care medicine 171, 899-907, doi:10.1164/rccm.200311-1535OC (2005).

Figure 1 H has an arrow in the SMA panel for the normal lung. Why is this there?

Thanks for pointing out this. The arrow indicates a positive internal control, positive staining of the blood vessel wall.

Figure 7 Legend is poorly written and difficult to understand. The first sentence is over 60 words in length and should be broken into at least two sentences if not more.

We thank the reviewer for this suggestion to improve style and divided the sentence in several shorter ones.

" Figure 7. Schematic diagram of the proposed mechanisms of fibrosis clearance. Left: In fibrotic lung, we find persistent myofibroblast activation in fibrotic plaques and JUN upregulation. JUN expression in fibrosis-associated fibroblasts (FAFs) appears to directly control the promoters and enhancers of CD47 and PD-L1. The direct consequence is increased expression of these immune checkpoint proteins on fibroblasts and dormant macrophages which do not phagocytose, but continue to release chronic inflammatory cytokines. JUN also directly regulates IL-6 at the chromatin level. The increased expression and secretion of this potent cytokine leads to a suppressive adaptive immune response—chiefly T cell exhaustion and upregulation of regulatory T cells. Right: Disrupting the suppression of the innate and adaptive immunity with CD47 and PD-L1 inhibitors as well as the proinflammatory IL-6 cytokine pathway stimulated phagocytic removal of profibrotic fibroblasts and T cell activation leading to clearance of the fibrosis in the lung."

Reviewer#4 (Remarks to the Author):

The Authors answered all my questions and from my side, I do not have further comments.

We thank reviewer #4 for his comments.

Reviewers' comments:

Reviewer #1 (Remarks to the Author):

Already in my previous peer review to the second version of the manuscript, I have expressed my positive opinion on the novelty and the quality of the work. I also recommended it for publication at Nature Communications. I maintain my previous opinion. Congratulations to the authors for the nice work.

Reviewer #2 (Remarks to the Author):

The manuscript by Cui et al continues to improve. The authors have been responsive to peer review and now present a concise story that will provide an excellent addition to the field. I have three concerns that I hope will be easy for the authors to address. No new experiments are required, however cleaning up two of the fluorescent photomicrographs will (as described below) will add to the paper.

1. Although some of the grammatical errors have been corrected, a number still remain. These will need to be corrected prior to publication. I don't know the best way to achieve this. Perhaps these could be corrected with the assistance of a manuscript editor. I defer to the Editor on this point.

2. Figure 1h. Remains a concern, as stated in the prior review. In their response letter, the authors provide more convincing data. I recommend using these newer images in the manuscript rather than the current images and placing the current images in the supplement. The current images continue to be plagued by several issues (as highlighted previously). First, CD47 staining is very dim and is not convincing. Second, collagen stains the tissues rather than cells and doesn't permit co-localization with fibroblasts (which is the point of the figure). Third, in the absence of single stained controls, how can the authors be sure that autofluorescence isn't an issue.

3. Figure 2.

a) The title for the figure legend is "Lung fibrotic condition converts macrophages into an immunosuppressive phenotype." I recommend a different title, since the figure doesn't show cause

and affect, merely an association. It might be possible that an immunosuppressive phenotype exists before the onset of fibrosis, rather than as a result.

b) In figure 2g, I'm concerned about CD68 staining in the normal lung. The CD68 positive areas are quite large and very bright. Moreover, many appear to lack Dapi stained nuclei. Therefore, I question whether these are truly CD68 positive cells. If the authors could provide a better image, it would provide reassurance.

Reviewers' comments:

Reviewer #1 (Remarks to the Author):

Already in my previous peer review to the second version of the manuscript, I have expressed my positive opinion on the novelty and the quality of the work. I also recommended it for publication at Nature Communications. I maintain my previous opinion. Congratulations to the authors for the nice work.

We thank reviewer #1 for his comments.

Reviewer #2 (Remarks to the Author):

The manuscript by Cui et al continues to improve. The authors have been responsive to peer review and now present a concise story that will provide an excellent addition to the field. I have three concerns that I hope will be easy for the authors to address. No new experiments are required, however cleaning up two of the fluorescent photomicrographs will (as described below) will add to the paper.

1. Although some of the grammatical errors have been corrected, a number still remain. These will need to be corrected prior to publication. I don't know the best way to achieve this. Perhaps these could be corrected with the assistance of a manuscript editor. I defer to the Editor on this point.

We are grateful to reviewer #2 recognizing the impact of our study, and for suggesting having our manuscript read by a native English speaker, which we did. All grammatical errors have been resolved by a professional native English editing service.

2. Figure 1h. Remains a concern, as stated in the prior review. In their response letter, the authors provide more convincing data. I recommend using these newer images in the manuscript rather than the current images and placing the current images in the supplement. The current images continue to be plagued by several issues (as highlighted previously). First, CD47 staining is very dim and is not convincing. Second, collagen stains the tissues rather than cells and doesn't permit co-localization with fibroblasts (which is the point of the figure). Third, in the absence of single stained controls, how can the authors be sure that autofluorescence isn't an issue.

We thank the reviewer for his suggestion to rearrange the images in Figure 1h. The immune stained images for CD47 have been replaced by the newer images.

3. Figure 2.

a) The title for the figure legend is "Lung fibrotic condition converts macrophages into an immunosuppressive phenotype." I recommend a different title, since the figure doesn't show cause and affect, merely an association. It might be possible that an immunosuppressive phenotype exists before the onset of fibrosis, rather than as a result.

We completely agree with the reviewer's comment. We only demonstrate the presence of immune suppressive subsets of macrophages in lung fibrosis at that given stage in disease and therefore suggest the following subtitle "Immune suppressive subsets of macrophages predominate in lung fibrosis"

b) In figure 2g, I'm concerned about CD68 staining in the normal lung. The CD68 positive areas are quite large and very bright. Moreover, many appear to lack Dapi stained nuclei. Therefore, I question whether these are truly CD68 positive cells. If the authors could provide a better image, it would provide reassurance.

We thank the reviewer and replaced the CD68 stained images for normal lung.